# Birds of a feather moult together: Differences in moulting distribution of four species of storm-petrels

**Anne N. M. A. Ausems**[1]*, **Grzegorz Skrzypek**[2], **Katarzyna Wojczulanis-Jakubas**[1], **Dariusz Jakubas**[1]

**1** Department of Vertebrate Ecology and Zoology, Faculty of Biology, The University of Gdańsk, Gdańsk, Poland, **2** West Australian Biogeochemistry Centre, The University of Western Australia, Crawley, WA, Australia

* anne.ausems@gmail.com

**Data Availability Statement:** All relevant data are within the manuscript and its Supporting Information files.

## Abstract

The non-breeding period of pelagic seabirds, and particularly the moulting stage, is an important, but understudied part of their annual cycle as they are hardly accessible outside of the breeding period. Knowledge about the moulting ecology of seabirds is important to understand the challenges they face outside and within the breeding season. Here, we combined stable carbon ($\delta^{13}$C) and oxygen ($\delta^{18}$O) signatures of rectrices grown during the non-breeding period of two pairs of storm-petrel species breeding in the northern (European storm-petrel, *Hydrobates pelagicus*, ESP; Leach's storm-petrel, *Hydrobates leucorhous*, LSP) and southern (black-bellied storm-petrel, *Fregetta tropica*, BBSP; Wilson's storm-petrel, *Oceanites oceanicus*, WSP) hemispheres to determine differences in moulting ranges within and between species. To understand clustering patterns in $\delta^{13}$C and $\delta^{18}$O moulting signatures, we examined various variables: species, sexes, years, morphologies (feather growth rate, body mass, tarsus length, wing length) and $\delta^{15}$N. We found that different factors could explain the differences within and between the four species. We additionally employed a geographical distribution prediction model based on oceanic $\delta^{13}$C and $\delta^{18}$O isoscapes, combined with chlorophyll-*a* concentrations and observational data to predict potential moulting areas of the sampled feather type. The northern species were predicted to moult in temperate and tropical Atlantic zones. BBSP was predicted to moult on the southern hemisphere north of the Southern Ocean, while WSP was predicted to moult further North, including in the Arctic and northern Pacific. While moulting distribution can only be estimated on large geographical scales using $\delta^{13}$C and $\delta^{18}$O, validating predictive outcomes with food availability proxies and observational data may provide valuable insights into important moulting grounds. Establishing those, in turn, is important for conservation management of elusive pelagic seabirds.

**Funding:** The study was funded by a grant of the National Science Centre, Poland to Dr. Dariusz Jakubas (2015/19/B/NZ8/01981; https://www.ncn.gov.pl/?language=en). The funders had no role in study design, data collection and analysis, decision to publish, or preparation of the manuscript.

**Competing interests:** The authors have declared that no competing interests exist.

## Introduction

The non-breeding period is an important part of the avian annual cycle, and it often spans the majority of the year in pelagic seabirds. Knowledge about seabird non-breeding ecology is crucial to understand the entire annual avian cycle as events on the non-breeding grounds may carry-over into the breeding period. Differences in non-breeding distribution and food availability [1], diet quality [2] or diet composition [3] at the non-breeding area may affect survival [4] and breeding success [5] during the subsequent breeding season. Additionally, contaminant accumulation during one stage of the annual cycle may be carried over to other stages, such as maternal transfer of contaminants to eggs and chicks [6, 7]. Nevertheless, the non-breeding period is understudied in many pelagic seabird species due to the inaccessibility of the birds beyond the breeding period.

An important stage in the non-breeding period of many pelagic seabird species is the moulting stage. Moulting is an energetically costly process [8], and plumage gaps caused by missing feathers increase flight costs through lowered flight efficiency [9], and reduced aerodynamic performance through lowered manoeuvrability [10, 11]. In pelagic seabirds moulting individuals may spend more time floating on the water than outside of the moulting period [12], affecting foraging effectiveness. Many pelagic seabirds, therefore, spread the impact of moult by reducing the number of feathers moulting at once [13], thus increasing the length of the moulting period. The extended moulting period thus covers a large part of the non-breeding period, and may even overlap with the end of the breeding period [14].

Studying the non-breeding distribution of small pelagic seabirds, such as storm-petrels, is still a challenge, resulting in a considerable knowledge gap. Although progress is being made with the miniaturisation of devices enabling whole year tracking [15–17], sample sizes remain relatively small due to low retrieval rates and incomplete tracks [15–17]. Additionally, these devices, while not proven detrimental [15–17], may be considered a relatively invasive method to study year-round movements.

Stable carbon isotope analysis in various avian tissues is a well-established method to determine seabird trophic level and foraging distribution during the moulting period [18, 19]. Stable isotope compositions of feathers remain inert after formation and thus represent the isotopic signatures of the prey eaten during feather synthesis [20], following the principle "you are what you eat" [21]. Since many seabird species complete their moulting during the non-breeding period, feather stable isotope analysis can be used to examine some of the ecological aspects of this part of the avian annual cycle.

Storm-petrels are typical pelagic seabirds, as they are highly mobile [15] and with feather growth taking up to several weeks [22]. Hence, isotope analysis applied to reconstruct birds' migratory movements provides a summary value for the feather growth period. However, combining multiple isotopes considerably increases the resolution to a more regional scale.

In this study, we aimed to characterise isotopic niches and use them to predict differences in moulting distribution of two species pairs of migratory storm-petrels, breeding sympatrically in both the northern (European storm-petrel, *Hydrobates pelagicus*, ESP; Leach's storm-petrel, *Hydrobates leucorhous*, LSP) and southern (black-bellied storm-petrel, *Fregetta tropica*, BBSP; Wilson's storm-petrel, *Oceanites oceanicus*, WSP) hemispheres. The latter species is considered the world's most abundant seabird species. However, relatively little is known about storm-petrel ecology during the non-breeding period. Due to their abundance, they may play an important role in global marine ecosystems, significantly influencing marine food webs. Additionally, due to their prevalence, and small size, they may be affected by anthropological disturbances and pollution differently than larger species. As such, they could be used as valuable sentinel species [23], but for that more knowledge is needed about their ecological niches during the non-breeding period.

To characterise stable isotopic niches and determine differences in ecological range during moult, we used the stable isotope composition of two elements: $\delta^{13}$C and $\delta^{18}$O of tail feathers moulted during the non-breeding period, largely simultaneously with other flight feathers [14, 24]. Stable isotope compositions of both elements vary spatially in marine ecosystems; $\delta^{13}$C values are correlated with phytoplankton distribution [25, 26] while $\delta^{18}$O values are correlated with salinity and fresh water input [27]. Both marine isotope values follow inshore/offshore gradients [26–28]. To our knowledge, this study is the first to combine $\delta^{13}$C and $\delta^{18}$O analyses to determine differences in moulting distributions of storm-petrels breeding sympatrically in both hemispheres. Traditionally, $\delta^{13}$C is combined with $\delta^{15}$N to study species' trophic and isotopic niches as nitrogen isotopic compositions serve as an important proxy for trophic level. However, this can only be used at local scales, e.g. during the breeding period when seabirds act as central place foragers having a restricted foraging range. During the non-breeding period pelagic seabirds roam freely through the vast oceans with spatially variable $\delta^{15}$N values. Specific predator trophic positions can only be inferred from bulk $\delta^{15}$N values if bulk $\delta^{15}$N values of lower trophic positions is known [29, 30].

As different moulting areas may vary in food availability, we tested whether moulting distribution differed in feather growth rate, a proxy for nutritional status during moulting [31]. Furthermore, we explored differences in feather $\delta^{15}$N between moulting niches as an additional, but cautious measurement for food availability and foraging location, as $\delta^{15}$N values are heavily dependent on trophic level, food source and foraging location [32]. Moreover, we expected that differences in stable isotopic niches and moulting distribution may be linked to differences in body size and sex within species. A study on several Procellariiformes species showed that in larger, size-dimorphic, species $\delta^{13}$C values in females were higher than in males, suggesting a more northerly distribution, while no isotopic differences were found in species not displaying sexual dimorphism [33]. In storm-petrels, sexual isotopic segregation was previously found in several species [34, 35], but not all [36].

To visualise and further interpret the differences in moulting niches, we used a predictive model to estimate moulting locations based on oceanic $\delta^{13}$C and $\delta^{18}$O gradients [37]. We verified these predictions using observations of storm-petrels recorded in online databases, and placed their predicted moulting grounds in established ecoregions [38]. Additionally, we used oceanic chlorophyll-*a* concentrations from the non-breeding period to validate predicted moult areas, as high chlorophyll-*a* concentrations (a proxy of high primary productivity) have been linked to areas with high seabird abundance [39], and highly productive marine areas are preferred moulting grounds [40].

## Materials and methods

### Study species and location

We captured ESP and LSP adults in August of 2018 (n = 52; n = 56, respectively) and 2019 (n = 40; n = 37, respectively) on the island of Mykines, Faroe Islands (62˚05´N, 07˚39´W), and BBSP and WSP adults during the austral summer of 2017 (n = 15; n = 100, respectively) and 2018 (n = 19; n = 126, respectively) around the Henryk Arctowski Polish Antarctic Station, on King George Island, South Shetland Islands, Antarctica (62˚09´S, 58˚27´W). ESP are the world's second smallest seabirds, while WSP is the smallest endotherms breeding on Antarctica. The northern (*Hydrobatidae*) and southern (*Oceanitidae*) species represent two different subfamilies [41, 42] differing in morphology and breeding ranges. The species name of LSP was therefore recently changed by BirdLife from *Oceanodroma leucorhoa* to *Hydrobates leucorhous* [43] though the old nomenclature is still widely used as well. BBSP and LSP have similar body sizes, except for tarsus length, and are larger than both ESP and WSP [44].

Both northern storm-petrel species have been observed along the west coast of Africa during their non-breeding season, generally as far south as the Cape of Good Hope [16, 17, 45], and LSP has been observed close to the Antarctic Peninsula [46]. A study based on stable $\delta^{13}$C and $\delta^{15}$N isotopes suggests that ESP from different Atlantic breeding colonies share moulting grounds as feathers grown during their non-breeding periods had similar stable isotope compositions. Contrastingly, feathers grown during their breeding periods had different stable isotope signatures [47]. A study on GLS-tracked LSP from Canadian colonies revealed that they moult in several geographically distinct areas [16]. WSPs can be observed around the British Isles [48, 49], though most are expected to moult south of the Subtropical Front [50]. The non-breeding distribution of BBSP is vastly understudied, but they are assumed not to cross the equator [44], though they have been sporadically observed in the North Atlantic [48], and likely moult in different oceanic zones than WSP [51].

The period from egg-laying to fledging takes on average 3.5 months in all four studied species [44, 52, 53]. All breeding activities take place during the summer (boreal and austral for the northern and southern hemisphere species, respectively), and though chicks generally fledge in late summer, in the northern hemisphere occasional late breeding attempts may be observed until autumn [54]. WSPs and BBSPs moult their flight feathers fully outside of the breeding period [24], while both ESPs and LSPs have been observed to overlap the start of flight feather moult with the last stages of chick-rearing [14, 55–57]. LSP rectrix moult overlaps more extensively with the breeding season than ESP rectrix moult [14, 55, 57]. Additionally, while in both species tail feather moult is irregular, ESPs seem to start tail moult with the central rectrix pair while LSP is more likely to start from the outer pair [14, 58]. However, the order in which tail feathers are moulted and the position of the start of tail moult is not conclusive [1/3 of observed LSP did not start tail moult at the outer rectrix pair; [14]].

## Data collection

**Field study.**   We captured adults of all studied species using mist-nets in the colony at night and by taking incubating BBSP and WSP birds from their nest. From each individual, we collected the right outer rectrix expected to be grown during the non-breeding season. For each sampled individual we measured body mass to the nearest 0.1 g using a digital scale (Pesola PTS3000, Switzerland), tarsus to the nearest 0.1 mm using callipers and folded wing length to the nearest 1 mm using a wing ruler. We determined the feather growth rate for the outermost rectrix by measuring growth bar width to the nearest 0.1 mm × d$^{-1}$. Feather growth bars are visible as alternating light and dark bands, formed during feather synthesis, but see Ausems et al. 2019 [59] for a detailed description of the method used.

It took several weeks for the sampled rectrices to be fully grown (ESP 30.6 ± 8.5 d; LSP 40.8 ± 13.0 d; BBSP 18.5 ± 2.5 d; WSP 18.7 ± 3.0 d; [59]). Thus, the rectrix formation period, overlapping to a considerable extent with flight feather moult [14, 24], includes a considerable part of the non-breeding period, even if feather growth started at the end of the breeding season. Although sampling tail-feathers increases the chance of sampling a feather moulted during the breeding period in LSP, we considered the uncertainty around the location of the start of tail moult too great to justify adding the negative effect of increasing feather gaps by sampling a more central feather.

**Molecular sexing.**   For molecular sexing, we collected several body feathers from the back of the neck from each individual from WSP and BBSP, and a drop of blood, stored in 70% ethanol, from ESP and LSP. We extracted DNA from the feathers and the blood after evaporation of the ethanol using the Sherlock AX (feathers) and Blood Mini kit (blood; A&A Biotechnology, Gdynia, Poland). We followed Griffiths et al. 1998 [60] to perform molecular sexing with

primer pair 2550F and 2718R but adapted the protocol by using 50˚C for the annealing temperature in the polymerase chain reaction (PCR). The primer pair amplifies introns on the CHD-W and CDH-Z genes located on the Wand Z avian sex chromosomes that vary in length [60]. The difference between the two fragments (~200 bp) was clearly visible in UV-light when separating on 2% agarose gel, stained with Midori Green. Some of the samples did not give reliable PCR products, thus for the southern species, we tested the sex of a total of 3 BBSP (2 females; 1 male) and 76 WSP (29 females; 47 males). In BBSP and WSP we additionally assigned sexes to 1 BBSP male, 7 WSP females and 2 WSP males based on the sex of the partner caught in the same nest. For the northern species, we successfully determined sex in 2018 and 2019 for 77 ESP (23 females; 54 males) and 52 LSP (9 females, 43 males).

**Ethics statement.** The Antarctic part of the study was conducted under the permission of the Polish National Standing Committee on Agricultural Research, Institute of Biochemistry and Biophysics (permit for entering the Antarctic Specially Protected Area No. 3/2016 & No. 08/2017). All birds captured on the Faroe Islands were handled under licenses of the Statens Naturhistoriske Museum, Københavns Universitet issued to AA (C 1012). All tissue samples on the Faroe Islands were taken with the permission of the Faroese Food and Veterinary Authority (19/01411-9) issued to AA. The study sites on the Faroe Islands was on privately owned land the local landowners gave permission to enter the study sites.

## Stable isotope analyses

Before analyses, all collected feather samples were washed in a 2:1 chloroform:methanol solution and twice in methanol, then air-dried for 24 h. The samples were then cut up into sub-millimetre sections using stainless steel scalpel blades. The $\delta^{15}$N and $\delta^{13}$C compositions were analysed using a continuous flow system consisting of a Delta V Plus mass spectrometer connected with a Thermo Flush 1112 Elemental Analyser via Conflo IV (Thermo-Finnigan/Germany; [61]). Raw values were reduced to the international scale using multi-point normalisation [62], based on international standards provided by IAEA: $\delta^{13}$C –NBS22, USGS24, NBS19, LSVEC [63]; and for $\delta^{15}$N –N1, N2, USGS32 and laboratory standards. Stable $\delta^{18}$O composition was analysed using a TC/EA coupled with Delta XL Mass Spectrometer in continues flow mode (Thermo-Fisher Scientific). The $\delta^{18}$O results were normalised to the VSMOW scale based on USGS42 and USGS43 and the equilibration method [64]. All $\delta^{13}$C results are reported in ‰ on VPDB, $\delta^{15}$N in ‰ on Air and $\delta^{18}$O in ‰ on VSMOW scale [62], with an external analytical uncertainty (one standard deviation) of 0.10 ‰ for $\delta^{13}$C and $\delta^{15}$N, and 0.50 ‰ for $\delta^{18}$O.

## Statistical analyses

All statistical analyses were done in R version 3.6.3. [65]. Individuals missing $\delta^{13}$C or $\delta^{18}$O values were removed from further analyses. From ESP two apparent outliers with $\delta^{13}$C < -23 ‰ were removed for further analyses, as we could not determine whether these values were caused by biological processes (i.e. different moulting ranges or ages) or were due to measurement errors. The results for the analyses including the outliers are reported in the (S2 File).

**Factors correlated with moult distribution differences.** We determined whether differences in $\delta^{13}$C and $\delta^{18}$O values were correlated with $\delta^{15}$N, feather growth rate, body mass, tarsus length, wing length, sex, and sample year with a conditional inference tree (CIT; function *ctree*; package *partykit*; [66, 67]). The CIT is a non-parametric regression tree, examining the relationship between multiple explanatory variables and one or multiple response variables. The *ctree* function estimates a regression relationship by binary recursive partitioning in a conditional inference framework. CIT outputs are in the form of an 'inverted tree', such that the

root at the top of the tree contains all observations, which is then divided into two branches, and again at each subsequent node. The aim of splitting the data at each step is to establish groups with a between-variation as large as possible and a within-variation as small as possible. Each node contains information about the explanatory variable name, its probability value, and the cut-off value in case of continuous explanatory variables [68]. CIT uses a machine learning algorithm to determine when splitting into further branches is no longer valid using a statistically determined stopping criterion; an *a priori* p-value [66]. CIT is robust against typical regression violations, such as over-fitting, (multi-) collinearity, and biases with regard to the types of explanatory variables used. To perform the CIT analysis we defined a multivariate response model for both $\delta^{13}$C and $\delta^{18}$O for both hemispheres separately adding species as an explanatory factor along with the aforementioned explanatory variables (i.e. $\delta^{13}$C + $\delta^{18}$O ~ $\delta^{15}$N + year + species + sex + feather growth rate + body mass + tarsus length + wing length).

We used a Welch's two sample t-tests (function *t.test*) to further explore the differences in $\delta^{13}$C and $\delta^{18}$O for each node. Additionally, for species with more than two terminal CIT nodes, we used a MANOVA (function *manova*) to further explore differences in $\delta^{13}$C and $\delta^{18}$O between terminal nodes, followed by a univariate ANOVA (function *aov*) when the MANOVA results were significant. Significant ANOVA results were followed by a Tukey HSD *post hoc* test (function *TukeyHSD*). Both $\delta^{13}$C and $\delta^{18}$O can be reasonably assumed to be normally distributed within the populations with homogenous variances, though the sample sizes within terminal CIT nodes were often too low to test these assumptions. We used the terminal nodes defined by the CIT model to group individuals for further analyses.

**Predicted moulting areas.** We created probability-of-origin raster maps for each storm-petrel species terminal CIT node (also called groups) based on both $\delta^{13}$C and $\delta^{18}$O signatures with the *isocat* package [69]. The probability-of-origin values produced by *isocat* range from 0 to 1, with low values indicating a low probability that the sample originated from that area, and high values indicating a high probability of origin (i.e. they should not be confused with *p*-values where a low value is generally preferred). Probability-of-origin values were calculated for each 1 × 1° (Latitude × Longitude) oceanic grid cell for $\delta^{13}$C and $\delta^{18}$O separately. As we did not necessarily expect each stable isotope to have similar probability-of-origin values for each cell, we summed the two values to generate one, combined probability-of-origin value per grid cell. The values presented here qualitatively, but not quantitatively, present the likelihood of bird presence during moulting within each species' subgroup partitioned using the CIT tree method. The expected primary spatial bird species distribution is in the regions > 95% quantile but these do not reflect bird population density. We used seasonally averaged plankton $\delta^{13}$C prediction isoscapes provided by C. Trueman from models described in Magozzi et al. 2017 [25], for the core non-breeding periods of the northern (November to March) and southern (May to October) species separately. For $\delta^{18}$O we used an annual averaged gridded dataset for Global Seawater Oxygen-18 Database isoscape obtained from LeGrande and Smith 2006 ([70]; https://data.giss.nasa.gov/o18data/) and visualized in ArcMap 10.3.1 [71]. For the two northern species, we only used data from the Atlantic Ocean as the studied populations do not migrate to other oceans and thus, we restricted the rasters to the area between 75°W and 52°E. For $\delta^{13}$C and $\delta^{18}$O isoscape maps see the (S1 File).

The calculated probability-of-origin values in all studied species differed by an order of magnitude (i.e. P × $10^{-5}$ for $\delta^{13}$C and P × $10^{-4}$ for $\delta^{18}$O). As we could not rule out this difference was due to artefacts caused by inappropriate discrimination factors, we centered and scaled both $\delta^{13}$C and $\delta^{18}$O probability-of-origin maps for each individual using the *scale* function (package *raster*; [72]) before summing the scaled probability-of-origin values in each cell. Before scaling, the probability-of-origin values were centered by subtracting the raster mean from each individual cell value. Scaling was then done by dividing the raster layers (i.e. all

probability-of-origin maps were grouped for both the northern and southern hemisphere separately) by their standard deviations. Due to the scaling procedure and summing the probability-of-origin values of both $\delta^{13}$C and $\delta^{18}$O, the probability-of-origin values reported in this study are thus factors reflecting probability and no longer range from 0 to 1. The reported probability-of-origin values are only meaningful in relation to the other probability-of-origin values within a hemisphere, such that a value can only be interpreted based on the distribution and range within each examined group (e.g. a value of 1 may be considered high if the values within the group range from -2 to 3, but may be considered low if the values range from 0 to 4). Therefore, in further analyses, values are compared to specific quantiles of the entire contemplated group to determine their meaning, and comparing values with values outside of the group is meaningless. We calculated the difference in predicted moult distribution maps using the Jaccard index (function *jaccard*; package *zonator*; [73]).

We corrected for differences in feather $\delta^{13}$C and $\delta^{18}$O compared to the source material (i.e. phytoplankton for $\delta^{13}$C and ocean water for $\delta^{18}$O) by subtracting trophic enrichment factors from the observed values. For seabirds, $\delta^{13}$C increases with trophic level, with trophic enrichment factors varying between species, sampled tissues and diet [74]. In storm-petrels, a trophic enrichment factor of 0.8 ‰ per trophic level has been used in previous studies [75, 76]. Although the exact trophic level of the studied storm-petrels during the non-breeding period is unknown, they consume mostly zooplankton [53, 77–87] and thus are at least two trophic levels higher than the source material. We, therefore, subtracted 1.6 ‰ from the observed $\delta^{13}$C values before starting the statistical analyses. As the discrimination factor between oceanic $\delta^{18}$O and feather $\delta^{18}$O were unknown, we calculated that based on 8 feathers known to be growing at the breeding site and the mean $\delta^{18}$O values of water samples taken within 5–120 km of the study site for each hemisphere. These rectrices were either actively growing when sampled or replaced a previously pulled feather (LSP n = 5, BBSP n = 1, WSP n = 2). We found discrimination factors of 10.4 ‰ & 13.0 ‰ between $\delta^{18}$O of ocean water and feathers grown during the breeding period, for the northern and southern species respectively.

**Moulting area verification.**   We validated the predicted moulting areas using chlorophyll-*a* concentrations as a proxy for food abundance which may serve as moulting areas [40], and observational data. We used chlorophyll-*a* concentrations at the surface layer from remote sensing MODIS Aqua satellite data (NASA Ocean Color Web, https://oceancolor.gsfc.nasa.gov/). We created concentration rasters for the corresponding core non-breeding periods for the species from the northern (November to March 2003–2018) and southern (May to October 2003–2018) hemispheres (S1). We averaged monthly maps in ArcMap 10.3.1 [71]. For the two northern species, we only used data from the Atlantic Ocean as the studied populations do not migrate to other oceans and thus restricted the rasters to the area between 75˚W and 52˚E. To find whether the predicted moulting areas were located in areas with increased primary productivity we firstly grouped the areas with lower 75% quantile of the scaled probability-of-origin values (i.e. 0–75% of the scaled probability-of-origin values) and the higher 25% quantile of the scaled probability-of-origin values (i.e. 76–100% of the scaled probability-of-origin values), and compared those two area categories using a Welch two sample *t*-test (function *t.test*).

For the southern species we also predicted the latitude at which the birds moulted for each terminal CIT node group using the equation from Quillfeldt et al. 2005 [76]: $\delta^{13}$C = -8.52 – (0.26 × latitude). We calculated the mean latitude for each terminal CIT node group, then extracted the scaled probability-of-origin values within the predicted latitude range (i.e. mean ± SD). If the predicted latitude was > -44˚, we referred to the estimated moulting area as north of the Subtropical Front, as the equation used is only accurate for predictions < -44˚ [50]. To determine the likelihood of the individuals moulting close to the predicted latitude we compared the maximum and mean ± standard deviation of the scaled probability-of-origin

values along the latitude range to the 95% quantile of scaled probability-of-origin values per terminal CIT node.

We grouped observations per species recorded in two online repositories from January 1990 until May 2020 [88, 89] in each 10° latitude × 10° longitude cell and calculated the average latitude and longitude for the observations within each cell. Not all species were consistently observed between years and areas due to differences in observation effort (e.g. the chance of observing an individual flying close to the shore in the Northern Atlantic is much higher than observing an individual flying in the pelagic waters of the Southern Atlantic due to an absence of observers), thus the observational data must be interpreted with caution.

We extracted the scaled probability-of-origin values within a buffer of $1.1 \times 10^6$ m (approximately 10°) around each observation point (average latitude and longitude) for each terminal CIT node group. For each terminal CIT node, we calculated the mean scaled probability-of-origin value per observation point, and the 50% and 95% quantiles for the entire map. We then compared the mean extracted scaled probability-of-origin value with the 50% and 95% quantile of the scaled probability-of-origin values for the whole raster map. Similarly, for each species, we extracted the chlorophyll-*a* concentration around each observation point and compared its mean with the 50% and 95% quantile of the chlorophyll-*a* raster for each hemisphere. Additionally, for each terminal CIT node for all species, we calculated the mean scaled probability-of-origin value in each marine eco-realm as defined in Spalding et al. 2007 [38], and compared those to the previously defined 50% and 95% scaled probability-of-origin quantiles.

## Results

### Stable isotopic moulting niches

The CIT for both hemispheres showed that carbon and oxygen isotopic signatures differed significantly between species (Node 1; p < 0.001; Fig 1, Table 1), but not between sexes. ESP had significantly higher $\delta^{13}$C values compared to LSP but lower $\delta^{18}$O values (Table 1), with $\delta^{18}$O being lower in 2019 than 2018 for both species and $\delta^{13}$C being lower in 2019 than 2018 for LSP but not differing for ESP (Table 2). Furthermore, within ESP individuals with tarsus length ≤ 23.5 mm differed significantly in moult distribution from individuals with tarsus length > 23.5 (Fig 1A), with individuals with shorter legs having significantly lower $\delta^{18}$O values than individuals with longer legs while $\delta^{13}$C did not differ (Table 1).

The CIT model for ESP including wing length and sex revealed confounding results, such that the significant dividing effect of tarsus length disappeared when including both wing length and sex, but not when including either one separately. Wing length is known to differ between sexes in storm-petrels [90, 91] and it was significantly longer in females than in males for LSP (Welch two-sample *t*-test; $t_{13.0} = 2.23$, p = 0.044) and WSP ($t_{57.2} = 2.63$; p = 0.011) in our study, but it did not differ significantly between males and females in ESP (Welch two-sample *t*-test: $t_{40.8} = 1.61$, p = 0.116; BBSP had too few sexed individuals to test). Therefore, we included either wing length or sex in the ESP CIT model. In both models, the CIT results were the same (Fig 1A), neither of which included wing length or sex.

The MANOVA results for the ESP terminal CIT nodes showed significant differences but this effect was only significant for $\delta^{18}$O and not for $\delta^{13}$C (Table 3). Terminal CIT node 3 had significantly higher $\delta^{18}$O values than terminal CIT node 5 (Tables 2 and 3).

In the southern hemisphere storm-petrel species the CIT revealed that BBSP had higher $\delta^{13}$C values and $\delta^{18}$O than WSP (Table 1). No further differences in moult distribution within BBSP were detected. In WSP nitrogen signatures significantly split studied birds into groups with $\delta^{15}$N values cut-off at 14.79 ‰ (Fig 1B); individuals with $\delta^{15}$N values lower or equal to the cut-off point had significantly lower $\delta^{13}$C values and higher $\delta^{18}$O values (Table 1) than

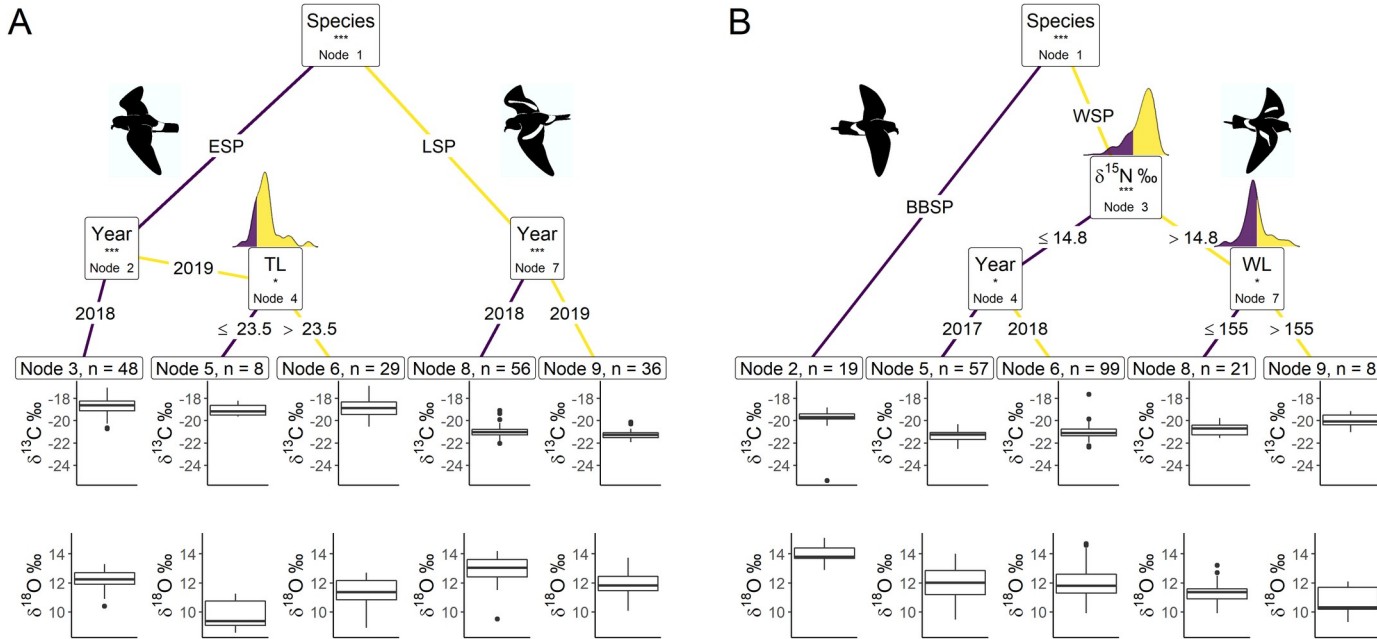

**Fig 1. Conditional inference trees characterising factors affecting the stable carbon ($\delta^{13}$C) and oxygen ($\delta^{18}$O) isotopic signatures.** We used Species, Year, $\delta^{15}$N (stable nitrogen isotope ratio), FGR (feather growth rate), BM (body mass), TL (tarsus length), and WL (wing length) as initial predictors. Body morphometrics (i.e. BM, TL and WL) were measured during the breeding season after moulting. Only variables with a significant dividing effect are shown in order of importance from the top down. At each node the dividing variable and corresponding p-value sign are listed in the box. These significance levels represent the test of independence between the listed variable and the response variables. Terminal CIT nodes indicate variable levels characterizing the response variable. Density plots above node boxes show the distribution of the continuous divisive variables, with the cut-off point dividing the colours. Boxplots show the median (band inside the box), the first (25%) and third (75%) quartile (box), the lowest and the highest values within 1.5 interquartile range (whiskers) and outliers (circles). (A) Northern hemisphere species; ESP–European storm-petrel; LSP–Leach's storm-petrel; (B) Southern hemisphere species; BBSP–black-bellied storm-petrel; WSP–Wilson's storm-petrel; n–number of individuals in each terminal CIT node group. P-values < 0.001 are shown with ***, p-values < 0.01 are shown with ** and p-values < 0.05 are shown with *.

individuals with higher $\delta^{15}$N values. The individuals with $\delta^{15}$N values ≤ 14.79 ‰ could be further split significantly by year (Fig 1B), with $\delta^{13}$C being significantly lower in 2017 than 2018, while $\delta^{18}$O did not differ (Table 1). The individuals with higher $\delta^{15}$N values could be further

**Table 1. Welch's two-sample *t*-test results for $\delta^{13}$C and $\delta^{18}$O values of the CIT internal nodes.**

| Hemisphere | Node | Variable | $\delta^{13}$C | | | $\delta^{18}$O | | |
|---|---|---|---|---|---|---|---|---|
| | | | df | t | p | df | t | p |
| Northern | 1 | Species | 139.8 | 22.4 | < **0.001** | 167.1 | -5.76 | **0.001** |
| | 2 | Year | 77.7 | 0.949 | 0.346 | 49.8 | 5.93 | < **0.001** |
| | 4 | Tarsus Length | 18.8 | -0.633 | 0.534 | 11.6 | -3.69 | **0.003** |
| | 7 | Year | 88.6 | 3.24 | **0.002** | 78.6 | 6.12 | < **0.001** |
| Southern | 1 | Species | 18.7 | 3.64 | **0.002** | 30.5 | 13.5 | < **0.001** |
| | 3 | $\delta^{15}$N | 37.0 | - 4.56, | < **0.001** | 42.2 | 4.20 | < **0.001** |
| | 4 | Year | 136.8 | -3.55 | < **0.001** | 102.7 | -0.104 | 0.917 |
| | 7 | Wing Length | 10.4 | -3.13 | **0.010** | 10.1 | 1.65 | 0.130 |

We tested the differences in stable carbon isotope ratios ($\delta^{13}$C) and stable oxygen isotope ratios ($\delta^{18}$O) between the child nodes of the conditional inference tree (CIT) analyses. Variable codes: Species, Year, $\delta^{15}$N (stable nitrogen isotope ratio), FGR (feather growth rate), BM (body mass), TL (tarsus length), and WL (wing length) as initial predictors. Body morphometrics (i.e. BM, TL and WL) were measured during the breeding season after moulting. Welch's two-sample *t*-test results: *df*–degrees of freedom; *t*–*t*- value. P-values < 0.05 are shown in **bold**.

**Table 2. Mean±SD $\delta^{13}$C and $\delta^{18}$O values of subgroups distinguished based on conditional inference tree terminal nodes.**

| Species | Terminal node | N | $\delta^{13}C_{VPDB}$ (‰) | $\delta^{18}O_{VSMOW}$ (‰) |
|---|---|---|---|---|
| ESP | 3 | 48 | -18.8 ± 0.8 | 12.2 ± 0.6 |
| | 5 | 8 | -19.1 ± 0.5 | 9.8 ± 1.0 |
| | 6 | 29 | -18.9 ± 0.8 | 11.3 ± 1.0 |
| | Total | 85 | -18.8 ± 0.8 (-20.7; -16.7) | 11.7 ± 1.1 (8.6; 13.3) |
| LSP | 8 | 56 | -21.0 ± 0.5 | 13.0 ± 0.8 |
| | 9 | 36 | -21.3 ± 0.4 | 11.9 ± 0.8 |
| | Total | 92 | -21.1 ± 0.5 (-22.0; -19.1) | 12.6 ± 1.0 (9.5; 14.2) |
| BBSP | Total | 19 | -19.9 ± 1.4 (-25.4; -18.8) | 14.0 ± 0.6 (12.9; 15.1) |
| WSP | 5 | 57 | -21.3 ± 0.5 | 12.0 ± 1.1 |
| | 6 | 99 | -21.0 ± 0.6 | 12.0 ± 0.9 |
| | 8 | 21 | -20.8 ± 0.5 | 11.4 ± 0.8 |
| | 9 | 8 | -20.0 ± 0.6 | 10.7 ± 1.0 |
| | Total | 185 | -21.1 ± 0.6 (-22.5; -17.6) | 11.9 ± 1.0 (9.3; 14.7) |

The species were split into groups with differing $\delta^{13}$C and $\delta^{18}$O values, based on variables described in the text. ESP–European storm-petrel; LSP–Leach's storm-petrel; BBSP–black-bellied storm-petrel; WSP–Wilson's storm-petrel; Terminal node–terminal CIT node number; n–sample size. Minimum and maximum values are provided at the species level in parentheses. See also Fig 1 for CIT results.

divided by wing length (Fig 1B). Individuals with wing lengths ≤ 155 mm had significantly lower $\delta^{13}$C values than individuals with wing lengths > 155 mm, while they did not differ in $\delta^{18}$O values (Table 1).

The WSP terminal CIT nodes groups differed significantly in both $\delta^{13}$C and $\delta^{18}$O (Table 3). Terminal CIT node 5 had significantly lower in $\delta^{13}$C values than terminal CIT node 8 and terminal CIT node 9 (Table 3). Terminal CIT node 5 $\delta^{18}$O values did not differ significantly from

**Table 3. Comparison of $\delta^{13}$C and $\delta^{18}$O values for species with > 2 CIT terminal nodes.**

| Species | MANOVA | | | SI | ANOVA | | | Tukey HSD | | |
|---|---|---|---|---|---|---|---|---|---|---|
| | df | F | p | | df | F | p | Pair | Dif | p |
| ESP | 2, 82 | 12.5 | < **0.001** | $\delta^{13}$C | 2, 82 | 0.56 | 0.572 | | | |
| | | | | $\delta^{18}$O | 2, 82 | 35.9 | < **0.001** | 5–3 | -2.45 | < **0.001** |
| | | | | | | | | 6–3 | -0.95 | < **0.001** |
| WSP | 3, 181 | 10.8 | < **0.001** | $\delta^{13}$C | 3, 181 | 16.5 | < **0.001** | 8–5 | 0.56 | < **0.001** |
| | | | | | | | | 9–5 | 1.34 | < **0.001** |
| | | | | | | | | 8–6 | 0.25 | 0.246 |
| | | | | | | | | 9–6 | 1.03 | < **0.001** |
| | | | | $\delta^{18}$O | 3, 181 | 5.95 | < **0.001** | 8–5 | -0.58 | 0.101 |
| | | | | | | | | 9–5 | -1.26 | **0.005** |
| | | | | | | | | 8–6 | -0.60 | 0.060 |
| | | | | | | | | 9–6 | -1.27 | **0.003** |

MANOVA followed by ANOVA and *post hoc* Tukey HSD tests were used to determine the differences in $\delta^{13}$C and $\delta^{18}$O between non-related terminal nodes. ESP–European storm-petrel; WSP–Wilson's storm-petrel; SI–tested stable isotope; df–degrees of freedom; F–F-value; pair–tested terminal node pair; dif–difference. P-values < 0.05 are in **bold**.

terminal CIT node 8 but did differ from terminal CIT node 9 (Table 3). Group 6 did not differ significantly from terminal CIT node 8, but did differ significantly from terminal CIT node 9 (Table 3).

## Predicted moult distribution

Based on CIT terminal nodes groups, ESP was split into three groups, LSP into two groups and WSP into four groups, differing in $\delta^{13}$C and $\delta^{18}$O (Table 2). BBSP was not split at all.

The similarity in scaled probability-of-origin distribution maps for ESP groups was very low (Jaccard index; terminal CIT node 3–5, J = 0.066; terminal CIT node 3–6, J = 0.041; terminal CIT node 5–6, J = 0.027). For LSP the similarity was higher than for ESP, but still relatively low (terminal CIT node 8–9, J = 0.181). WSP terminal CIT node 5 and 6 were nearly identical (J = 0.957), terminal CIT node 8 and 9 were fairly similar (J = 0.613). WSP terminal CIT node group 5 and 8, and group 6 and 8 shared approximately half of the same probability-of-origin value distributions (J = 0.502; J = 0.485, respectively), while terminal CIT node group 5 and 9, and group 6 and 9 shared approximately one-third of the scaled probability-of-origin value distributions (J = 0.299; J = 0.288, respectively). As BBSP was not separated into different terminal nodes, and thus did not have multiple scaled probability-of-origin maps, we did not calculate a Jaccard index.

Within the northern hemisphere species, we found significantly lower chlorophyll-$a$ concentrations in the areas with the 76%–100% highest scaled probability-of-origin values than in the lower 0%–75% value areas for all groups for all terminal CIT node groups (t-test; ESP group 3 $t_{215.9}$ = 9.32, p < 0.001; ESP group 6 $t_{255.1}$ = 3.05, p = 0.003; LSP group 8 $t_{230.5}$ = 9.80, p < 0.001; LSP group 9 $t_{221.9}$ = 6.90, p < 0.001; Figs 2 and 3) except for ESP group 5 ($t_{244.5}$ = -6.68, p < 0.001; Fig 2). For BBSP we found no significant difference in chlorophyll-$a$ concentrations between higher and lower scaled probability-of-origin areas ($t_{841.2}$ = 1.36, p = 0.174; Fig 4). In WSP we found significantly higher chlorophyll-$a$ concentrations in the areas with the 76%–100% highest scaled probability-of-origin values than in the lower 0%–75% value areas for all groups for all four terminal CIT nodes (group 5 $t_{615.2}$ = -15.8, p < 0.001; group 6 $t_{614.0}$ = -15.4, p < 0.001; group 8 $t_{621.1}$ = -17.0, p < 0.001; group 9 $t_{643.8}$ = -12.1, p < 0.001; Fig 5).

BBSP and WSP individuals in terminal CIT node group 9 were predicted to moult north of —44˚ (Table 4), and we thus did not analyse scaled probability-of-origin values around their predicted moulting latitudes. The mean ± SD of the scaled probability-of-origin values around the predicted moulting latitude for the other three WSP terminal CIT nodes were lower than the 95% quantile of the entire scaled probability-of-origin maps, although the maximum values of WSP terminal CIT nodes 5 and 6 were higher (Table 4; Fig 5).

In none of the marine eco-realms mean scaled probability-of-origin values for ESP individuals from terminal CIT node 3 were higher than the 95% quantile of the entire considered area (Tables 5 and 6; Fig 2; though only the Southern Ocean and the Arctic had mean scaled probability-of-origin values lower than the 50% quantile. Individuals from ESP terminal CIT node 5 had higher than the 95% quantile scaled probability-of-origin values for the Temperate Southern America eco-realm, while individuals from ESP terminal CIT node 6 had mean scaled probability-of-origin values higher than the 95% quantile for Temperate Southern Africa (Tables 5 and 6; Fig 2). In LSP neither terminal CIT nodes had mean scaled probability-of-origin values higher than the 95% quantile for any of the eco-realms. However, all eco-realms besides the Southern Ocean and Arctic had mean values higher than the 50% quantile (Tables 5 and 6; Fig 3). Similarly, for BBSP no eco-realms had mean scaled probability-of-origin values higher than the 95% quantile, but the Temperate Southern Africa, Western Indo-Pacific,

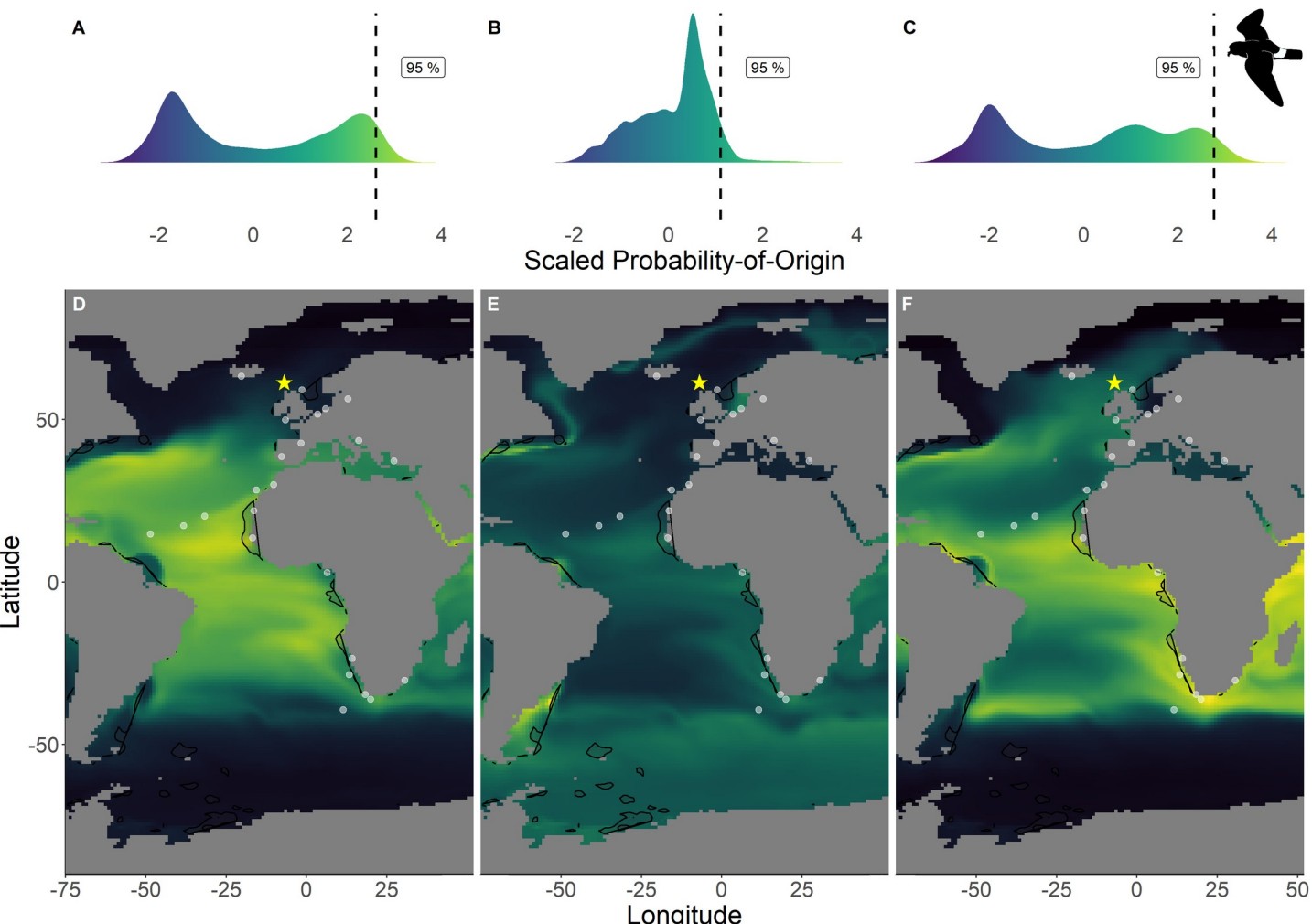

**Fig 2. Scaled probability-of-origin maps based on δ¹³C and δ¹⁸O for each group for the European storm-petrel.** Terminal nodes from a conditional inference tree (CIT) based on differences between years, and correlated to body morphology (Fig 1A) were treated as groups. (A) Scaled probability-of-origin value distribution for terminal CIT node 3; (B) scaled probability-of-origin value distribution for terminal CIT node 5; (C) scaled probability-of-origin value distribution for terminal CIT node 6; (D) scaled probability-of-origin map for terminal CIT node 3; (E) scaled probability-of-origin map for terminal CIT node 5; (F) scaled probability-of-origin map for terminal CIT node 6. Scaled probability-of-origin values are shown on a relative high (yellow)–low (black) gradient in both the density plots and maps. The 95% quantile of the scaled probability-of-origin values per terminal CIT node are shown with the dashed line. Shaded contours show high chlorophyll-*a* concentration areas (upper 95% of the data), and white dots show observation locations [88, 89]. The yellow star indicates the location of the breeding colony where birds were sampled.

Tropical Atlantic, Eastern Indo-Pacific, Central Indo-Pacific, Temperate Australasia and Tropical Eastern Pacific eco-realms had mean scaled probability-of-origin values higher than the 50% quantile (Tables 5 and 6; Fig 4). WSP terminal CIT nodes 5, 6 and 8 had mean scaled probability-of-origin values higher than the respective 95% quantiles for the Temperate Northern Pacific eco-realm (Tables 5 and 6; Fig 5). For terminal CIT nodes 5 and 6 the Southern Ocean, Temperate Northern Atlantic and Temperate Australasia eco-realms had mean scaled probability-of-origin values lower than the 50% quantiles, while the remaining eco-realms had mean scaled probability-of-origin values between the 50% and 95% quantiles (Tables 5 and 6, Fig 5). For WSP terminal CIT node 8 Temperate Australasia and the Southern Ocean had mean scaled probability-of-origin values lower than the 50% quantiles (Tables 5 and 6, Fig 5). WSP terminal CIT node 9 had mean scaled probability-of-origin values higher than the 95% quantile for the Arctic and Temperate Northern Pacific eco-realms (Tables 5 and 6, Fig 5). The

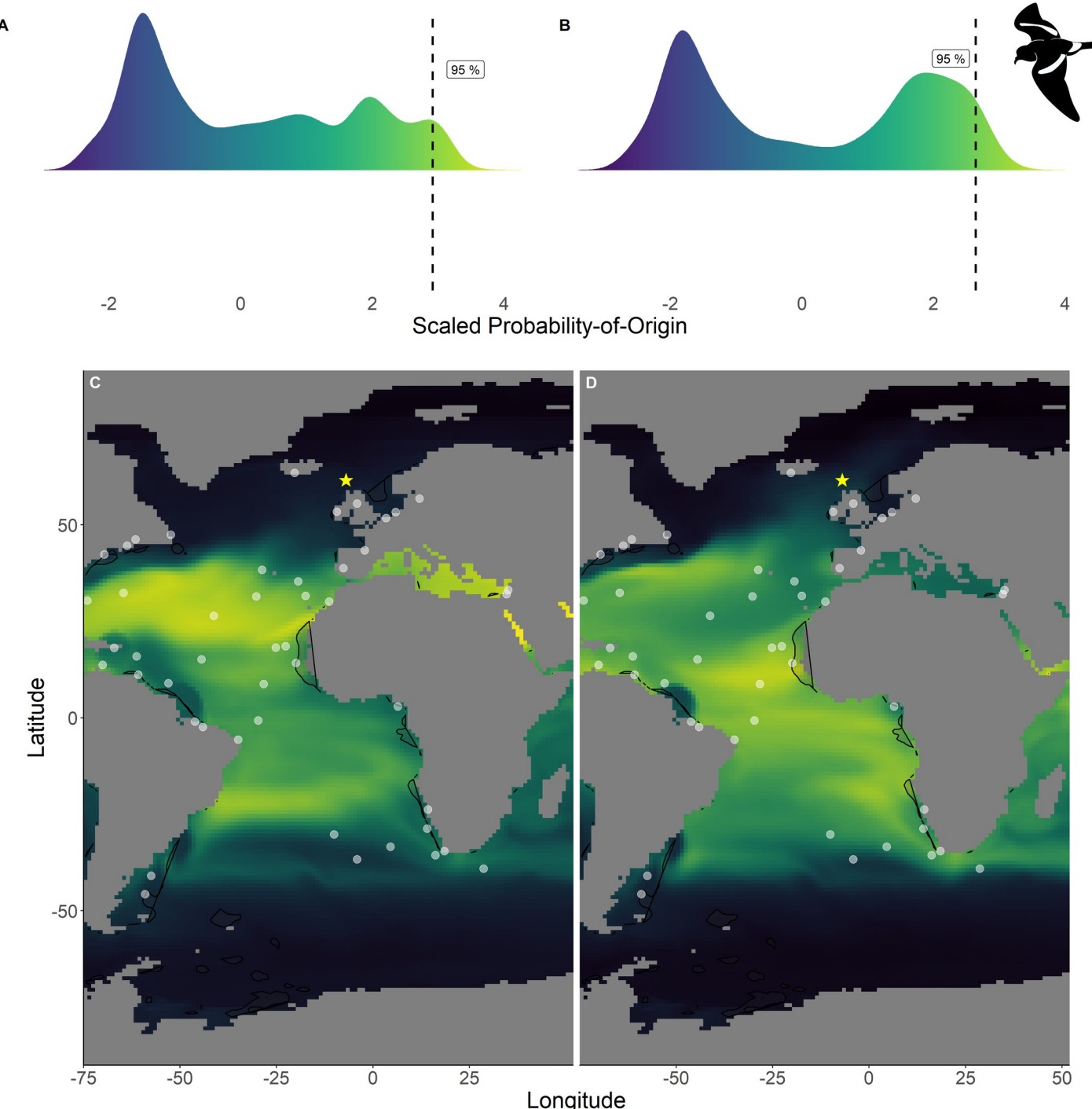

**Fig 3. Scaled probability-of-origin maps based on δ¹³C and δ¹⁸O for each group of the Leach's storm-petrel.** Terminal nodes from a conditional inference tree (CIT) based on differences between years, and correlated to body morphology (Fig 1A) were treated as groups. (A) Scaled probability-of-origin value distribution for terminal CIT node 8; (B) scaled probability-of-origin value distribution for terminal CIT node 9; (C) scaled probability-of-origin map for terminal CIT node 8; (D) scaled probability-of-origin map for terminal CIT node 9. Scaled probability-of-origin values are shown on a relative high (yellow)–low (black) gradient in both the density plots and maps. The 95% quantile of the scaled probability-of-origin values per terminal CIT node are shown with the dashed line. Shaded contours show high chlorophyll-*a* concentration areas (upper 95% of the data), and white dots show observation locations [88, 89]. The yellow star indicates the location of the breeding colony where birds were sampled.

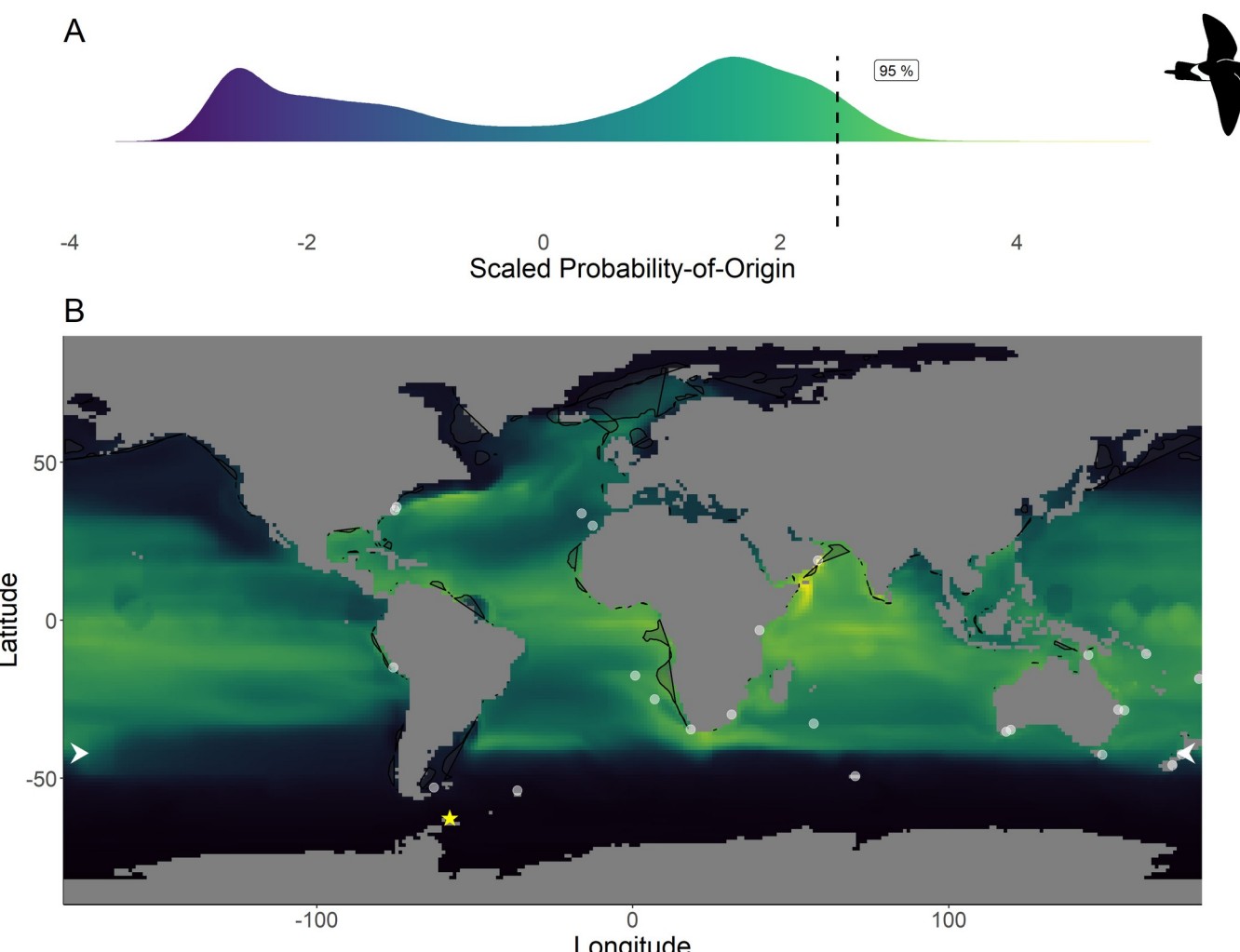

**Fig 4. Scaled probability-of-origin map based on $\delta^{13}$C and $\delta^{18}$O for the black-bellied storm-petrel.** (A) Scaled probability-of-origin value distribution; (B) scaled probability-of-origin map. Scaled probability-of-origin values are shown on a relative high (yellow)–low (black) gradient in both the density plot and map. The 95% quantile of the scaled probability-of-origin values is shown with the dashed line. Shaded contours show high chlorophyll-*a* concentration areas (upper 95% of the data), and white dots show observation locations [88, 89]. The white arrows at the edge of the plot show the predicted moulting latitude based on the equation from Quillfeldt et al. 2005([76]; Table 4). The yellow star indicates the location of the breeding colony where birds were sampled.

Temperate South America, Temperate Australasia and Southern Ocean eco-realm mean scaled probability-of-origin values for WSP terminal CIT node 9 were lower than the 50% quantiles, while the remaining eco-realms had mean scaled probability-of-origin values between the 50% and 95% quantiles (Tables 5 and 6, Fig 5).

## Observational data

The locations at which the species were observed during the non-breeding period were generally outside of the 95% quantile of the scaled probability-of-origin areas, although for all species except BBSP they were on average observed in the 75% quantile of the scaled probability-of-origin areas (Table 5). For all species the mean ± SD of the chlorophyll-*a* concentrations around the observation locations overlapped with the 95% quantile of the chlorophyll-*a* concentration for the entire area in which the species could be expected to moult, with the ESP mean being closest to the 95% quantile for the northern hemisphere species (Table 5).

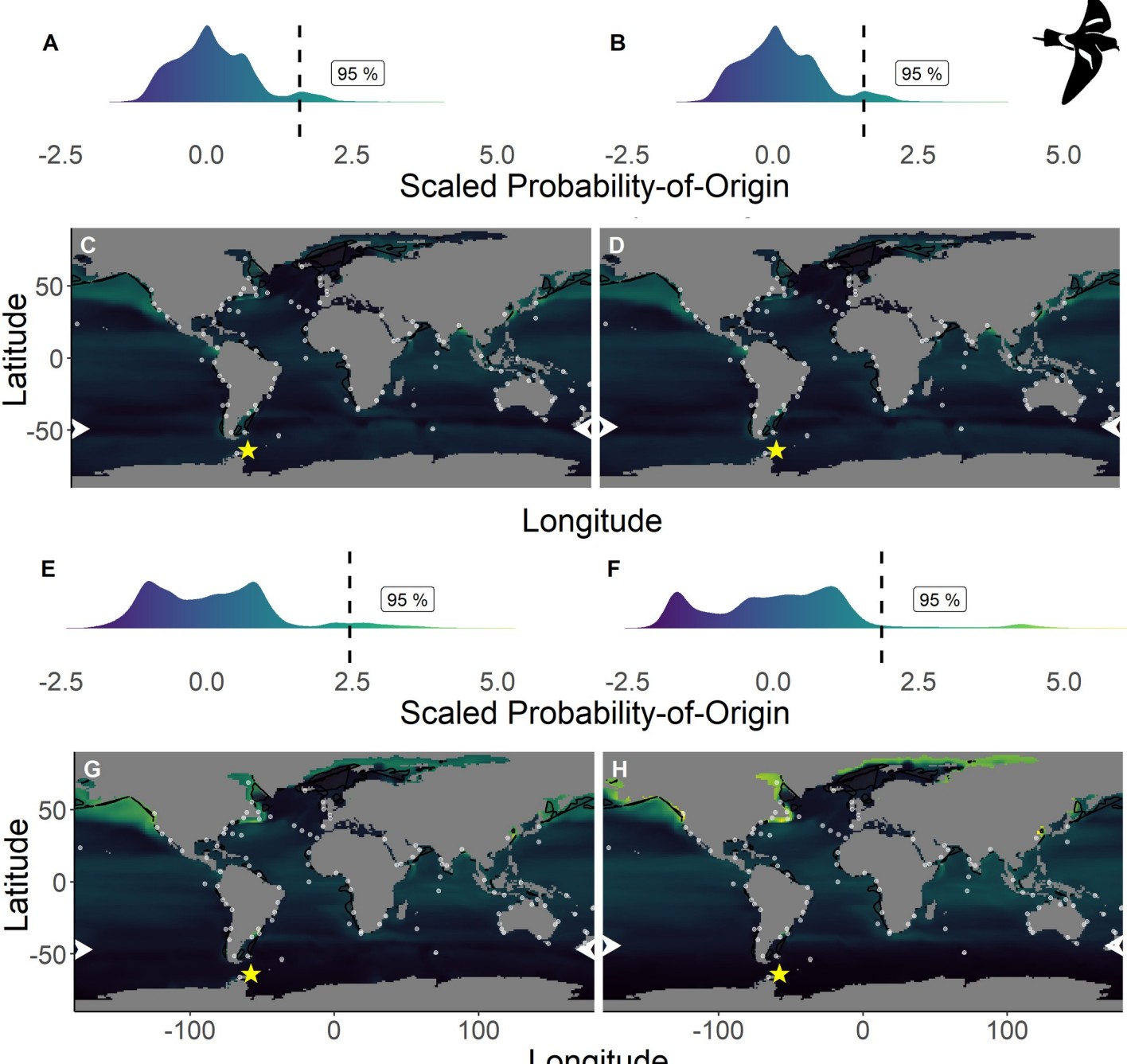

**Fig 5. Scaled probability-of-origin maps based on $\delta^{13}C$ and $\delta^{18}O$ for each group for the Wilson's storm-petrel.** Terminal nodes from a conditional inference tree (CIT) based on differences between years, and correlated to body morphology (Fig 1B) were treated as groups. (A) Scaled probability-of-origin value distribution for terminal CIT node 5; (B) scaled probability-of-origin value distribution for terminal CIT node 6; (C) scaled probability-of-origin map for terminal CIT node 5; (D) scaled probability-of-origin map for terminal CIT node 6; (E) scaled probability-of-origin value distribution for terminal CIT node 8; (F) scaled probability-of-origin value distribution for terminal CIT node 9; (G) scaled probability-of-origin map for terminal CIT node 8; (H) scaled probability-of-origin map for terminal CIT node 9. Scaled probability-of-origin values are shown on a relative high (yellow)–low (black) gradient in both the density plots and maps. The 95% quantile of the scaled probability-of-origin values per terminal CIT node are shown with the dashed line. Shaded contours show high chlorophyll-*a* concentration areas (upper 95% of the data), and white dots show observation locations [88, 89]. The white arrows at the edge of the plot show the predicted moulting latitude based on the equation from Quillfeldt et al. 2005([76]; Table 4). The yellow star indicates the location of the breeding colony where birds were sampled.

**Table 4. Scaled probability-of-origin values in the predicted moulting latitude range for the southern hemisphere species.**

| Species | Terminal Node | Latitude (˚) | Scaled probability-of-origin | | |
| --- | --- | --- | --- | --- | --- |
| | | | Maximum | Mean ± SD | 95% Quantile |
| BBSP | 2 | -41.3 ± 5.4 | NA | NA | NA |
| WSP | 5 | -47.1 ± 1.8 | 2.17 | -0.32 ± 0.60 | 1.57 |
| | 6 | -46.0 ± 2.2 | 2.14 | -0.37 ± 0.58 | 1.53 |
| | 8 | -45.0 ± 1.8 | 1.69 | -0.74 ± 0.29 | 2.45 |
| | 9 | -42.3 ± 2.4 | NA | NA | NA |

The latitude at which the species were expected to moult was predicted using the equation from Quillfeldt et al. 2005 [76]. Predicted maximum, mean and SD of the scaled probability-of-origin values were extracted from the SD latitude wide buffer around the predicted mean latitude for each terminal CIT node group. The 95% quantile of the scaled probability-of-origin values was calculated for the entire probability-of-origin map. We did not compare scaled probability-of-origin values for groups predicted to moult at < 44˚S as the equation used was not accurate further north (41). Terminal node–terminal CIT node; Latitude–predicted moulting latitude; BBSP–black-bellied storm-petrel; WSP–Wilson's storm-petrel. See also Figs 4 and 5 for the scaled probability-of-origin distributions. Note: the scaled probability-of-origin values are relative, i.e. not comparable between species from both hemispheres.

## Discussion

Our study combining stable carbon and oxygen isotopes ($\delta^{13}$C and $\delta^{18}$O), revealed that for both hemispheres the storm-petrel species moulted their rectrices in different areas, as they differed significantly in both isotopic signatures (Fig 1). ESP had significantly higher $\delta^{13}$C values and lower $\delta^{18}$O values than LSP, suggesting less pelagic moulting ranges for the former [92], while BBSP had both higher $\delta^{13}$C and higher $\delta^{18}$O values than WSP indicating a more pelagic lifestyle and moulting grounds further north for BBSP [76, 92]. The lack of any division into subgroups in BBSP is likely due to the relatively small sample size (n = 19). However, BBSP also generally has a more pelagic lifestyle [44, 52] with larger foraging areas where stable isotope ratios differ over larger areas, and as such the feather isotopic signatures may vary less between individuals as they are averaged over a wider range of sources.

**Table 5. Scaled probability-of-origin and chlorophyll-*a* concentration values around each observation location per terminal CIT node.**

| Species | Terminal node | Scaled probability-of-origin | | | Chlorophyll-*a* | | |
| --- | --- | --- | --- | --- | --- | --- | --- |
| | | Mean ± SD | 50% | 95% | Mean ± SD | 50% | 95% |
| ESP | 3 | 1.85 ± 0.38 | -0.17 | 2.60 | 1.33 ± 4.23 | 0.24 | 1.34 |
| | 5 | 0.75 ± 0.32 | 0.37 | 1.11 | | | |
| | 6 | 1.93 ± 0.60 | 0.45 | 2.76 | | | |
| LSP | 8 | 0.51 ± 0.32 | -0.19 | 2.92 | 0.55 ± 2.23 | | |
| | 9 | 0.52 ± 0.37 | -0.06 | 2.64 | | | |
| BBSP | 2 | 0.02 ± 0.51 | 0.76 | 2.48 | 0.96 ± 3.43 | 0.17 | 1.45 |
| WSP | 5 | -0.30 ± 0.47 | 0.02 | 1.57 | 0.78 ± 2.55 | | |
| | 6 | -0.26 ± 0.43 | 0.03 | 1.53 | | | |
| | 8 | -0.26 ± 0.46 | 0.04 | 2.45 | | | |
| | 9 | -0.29 ± 0.55 | 0.15 | 1.86 | | | |

Scaled probability-of-origin and chlorophyll-*a* concentration values were averaged for a buffer of approximately 10˚ around the average latitude and longitude for each observation location. The 50% and 95% quantiles were calculated for the entire raster, for both the scaled probability-of-origin maps and the chlorophyll-a concentration maps. ESP–European storm-petrel; LSP–Leach's storm-petrel; BBSP–black-bellied storm-petrel; WSP–Wilson's storm-petrel; Terminal node–terminal CIT node. See also Figs 2–5 for scaled probability-of-origin distributions. Note: the scaled probability-of-origin values are relative, i.e. not comparable between species from both hemispheres.

**Table 6. Mean±SD of the scaled probability-of-origin values per marine eco-realm per terminal CIT node.**

| | Species | | | | | | | | | |
|---|---|---|---|---|---|---|---|---|---|---|
| | ESP | | | LSP | | BBSP | WSP | | | |
| | Terminal node | | | | | | | | | |
| Eco-realm | 3 | 5 | 6 | 8 | 9 | 2 | 5 | 6 | 8 | 9 |
| Arctic | -1.86 ± 0.16 | -0.77 ± 0.50 | -2.07 ± 0.44 | -1.69 ± 0.16 | -1.94 ± 0.18 | -1.65 ± 0.34 | *0.45 ± 0.54* | *0.40 ± 0.53* | *1.87 ± 0.78* | **2.50 ± 1.13** |
| Central Indo-Pacific | NA ± NA | NA ± NA | NA ± NA | NA ± NA | NA ± NA | 1.85 ± 0.15 | 0.63 ± 0.14 | 0.64 ± 0.15 | 0.85 ± 0.13 | 1.10 ± 0.13 |
| Eastern Indo-Pacific | NA ± NA | NA ± NA | NA ± NA | NA ± NA | NA ± NA | 1.97 ± 0.17 | 0.15 ± 0.15 | 0.14 ± 0.15 | 0.42 ± 0.15 | 0.62 ± 0.15 |
| Southern Ocean | -1.58 ± 0.11 | *0.68 ± 0.11* | -1.85 ± 0.12 | -1.35 ± 0.12 | -1.68 ± 0.12 | -2.35 ± 0.16 | -0.11 ± 0.21 | -0.09 ± 0.21 | -1.01 ± 0.25 | -1.45 ± 0.15 |
| Temperate Australasia | NA ± NA | NA ± NA | NA ± NA | NA ± NA | NA ± NA | 1.32 ± 0.43 | -0.36 ± 0.13 | -0.37 ± 0.13 | -0.12 ± 0.15 | 0.08 ± 0.15 |
| Temperate Northern Atlantic | *0.36 ± 0.28* | -0.50 ± 0.48 | *0.07 ± 0.42* | *0.76 ± 0.34* | *0.27 ± 0.28* | 0.28 ± 0.47 | -0.12 ± 0.47 | -0.14 ± 0.47 | 0.42 ± 0.51 | 0.63 ± 0.71 |
| Temperate Northern Pacific | NA ± NA | NA ± NA | NA ± NA | NA ± NA | NA ± NA | -0.52 ± 0.26 | **1.61 ± 0.41** | *1.58 ± 0.41* | *2.28 ± 0.50* | **2.00 ± 0.61** |
| Temperate South America | -0.04 ± 0.28 | **1.19 ± 0.41** | *0.02 ± 0.43* | *0.11 ± 0.26* | -0.07 ± 0.31 | -0.5 ± 0.46 | *0.48 ± 0.55* | *0.47 ± 0.54* | *0.33 ± 0.71* | *0.09 ± 0.54* |
| Temperate Southern Africa | *1.61 ± 0.32* | *0.80 ± 0.15* | **2.88 ± 0.17** | *1.08 ± 0.25* | *1.98 ± 0.27* | 2.26 ± 0.24 | 0.29 ± 0.18 | 0.29 ± 0.18 | 0.57 ± 0.18 | 0.78 ± 0.18 |
| Tropical Atlantic | *2.17 ± 0.31* | *0.41 ± 0.23* | *2.09 ± 0.33* | *1.76 ± 0.27* | *2.25 ± 0.33* | 2.00 ± 0.30 | 0.37 ± 0.14 | 0.36 ± 0.15 | 0.68 ± 0.12 | 0.82 ± 0.12 |
| Tropical Eastern Pacific | NA ± NA | NA ± NA | NA ± NA | NA ± NA | NA ± NA | 0.83 ± 0.27 | 0.86 ± 0.38 | 0.88 ± 0.39 | 0.78 ± 0.25 | 0.96 ± 0.18 |
| Western Indo-Pacific | NA ± NA | NA ± NA | NA ± NA | NA ± NA | NA ± NA | 2.11 ± 0.25 | 0.75 ± 0.25 | 0.75 ± 0.25 | 0.98 ± 0.25 | 1.17 ± 0.21 |

Marine eco-realms were defined in Spalding et al. 2007 [38]. Mean scaled probability-of-origin values were compared to the respective 50% and 95% scaled probability-of-origin quantiles per terminal CIT node per species (Table 5). ESP–European storm-petrel; LSP–Leach's storm-petrel; BBSP–black-bellied storm-petrel; WSP–Wilson's storm-petrel; Terminal node–terminal CIT node. Mean scaled probability-of-origin values > 50% and < 95% quantiles of the corresponding terminal CIT node are italicised and mean scaled probability-of-origin values > 95% quantile of the corresponding terminal CIT node are **bolded**. NA–no data available as it was outside of the species distribution extent. See also Figs 2–5 for scaled probability-of-origin distributions. Note: the scaled probability-of-origin values are relative, i.e. not comparable between species from both hemispheres.

## Factors affecting moulting distribution

As the breeding seasons of the northern and southern species are at opposite times of the year, and fieldwork in the two hemispheres was carried out sequentially, the interannual difference in $\delta^{13}C$ and $\delta^{18}O$ differed for both hemispheres and that not necessarily had to be related to species characteristics. For example, an El Niño event was observed between the two studied breeding seasons in the northern hemisphere (2018–2019) but not between the two studied breeding seasons in the southern hemisphere (2017–2018) (e.g. https://www.climate.gov/news-features/blogs/enso/february-2019-enso-update-el-ni%C3%B1o-conditions-are-here, accessed 04-09-2020). El Niño events in the Pacific Ocean generally lead to lower sea surface temperatures and more stable weather conditions in the Atlantic Ocean [93]. Nevertheless, interannual variability in weather conditions is higher for the North Atlantic than the South, and in the first half of the year compared to the second [93]. As such, a stronger interannual variability in $\delta^{13}C$ and $\delta^{18}O$ during the non-breeding period could be expected for the studied northern hemisphere species than for the southern hemisphere species [28]. Indeed, year was only a significant factor in the CIT dividing WSP into groups predicted to have moulting ranges extending further South (Table 5).

For WSP $\delta^{15}N$ was the first dividing factor in the CIT (Fig 1B). Individuals from the lower $\delta^{15}N$ group had lower $\delta^{13}C$ but higher $\delta^{18}O$ values (Table 2) and were predicted to moult further South compared to individuals higher $\delta^{15}N$ group (Table 4; Fig 5). As $\delta^{15}N$ is linked both to trophic level [94] and foraging location [28] at wide geographical scales, we could not distinguish whether these differences were due to differences in foraging range, diet or a combination of both. However, areas around Alaska, Nova Scotia and the Labrador Sea show relatively high plankton $\delta^{15}N$ values [28, 92], and were also highlighted as high scaled probability-of-

origin areas for these two moult groups, after correcting for trophic enrichment factors (Fig 5). Additionally, WSP has been observed in high quantities close to some of these areas [88, 89], and individuals breeding on King George Island have been predicted to moult north of the Subtropical Front before [50, 76]. The Temperate Northern Pacific eco-realm was characterized by high scaled probability-of-origin values for all four groups of WSP distinguished in our study; for one of the WSP groups with high $\delta^{15}$N values the Arctic was designated as an important moulting region as well (Table 6). For both high $\delta^{15}$N values WSP groups (8 and 9), the temperate Northern Atlantic was shown to have relatively high scaled probability-of-origin values, but this region had relatively low scaled probability-of-origin values in the other two terminal CIT nodes (5 and 6; Table 6). Therefore, although $\delta^{15}$N may still be affected by differences in trophic level, we assume at least part of the variation in this variable may be explained by differences in moult distribution.

We found some morphological differences between individuals differing in moult distributions for ESP and WSP. These differences may be caused by a trade-off between foraging ability and flight costs during migration [95], and may be linked to sexual dimorphism [33]. In ESP we found that individuals with shorter tarsi had lower $\delta^{18}$O values than individuals with longer tarsi (Fig 1A; Table 2), and in WSP we found that individuals with shorter wings had significantly lower $\delta^{13}$C values than individuals with longer wings (Fig 1B; Table 2). As $\delta^{18}$O values in coastal zones close to large river mouths are lower compared to the open ocean due to increased freshwater input [27], this may indicate that ESP individuals with shorter legs forage closer to estuaries such as the Banc d'Arguin [96]. Additionally, $\delta^{18}$O values differ over a latitudinal gradient with $\delta^{18}$O being considerably lower closer to the polar regions than the Equator and tropical zones. ESP differs in migratory behaviour between sexes [97], and shows sexual dimorphism for several body measurements but not tarsus length [90]. WSP females are slightly larger than males [98], with wings being approximately 2.8% longer in females than males [91]. However, we did not find a significant effect of sex in CIT analyses.

Stable isotopic signatures of secondary feathers (S8) of the Monteiro's storm-petrel (*Hydrobates monteiroi*) moulted during the previous non-breeding period showed an evident isotopic niche segregation between sexes. Males exhibited higher $\delta^{13}$C and $\delta^{15}$N values, and larger isotopic niches compared to females, presumably caused by spatial sexual segregation and exploitation of areas of contrasting environmental conditions [34]. In the Madeiran storm-petrel (*Hydrobates castro*) $\delta^{15}$N values differed between sexes during the non-breeding period, with females having lower $\delta^{15}$N values than males, possibly caused by intersexual differences in distribution during the non-breeding season, or as a result of differences in diet between sexes or differences in the relative amount of different prey taken [35]. However, in Canadian LSP populations, no sexual isotopic segregation was found [86]. Our results thus suggest that the different moulting distributions are probably not caused by sexual segregation in ESP, LSP or WSP.

Thus, the dividing effect of tarsus length and wing length in ESP and WSP may be correlated with differences in foraging behaviour between individuals, rather than sexual segregation during the non-breeding period. Storm-petrel species with shorter tarsi show less pattering behaviour [99], and species with shorter wings are better adapted to the strong winds in polar regions to exploit less mobile, highly abundant prey [100]. Additionally, these effects were only present in a part of the studied individuals, indicating that differences in behaviour only arise under specific circumstances such as differences in prey availability between areas or years [101].

## Predicted moulting areas

Chlorophyll-*a* concentration being a proxy for primary production may be used to locate seabirds foraging hotspots [39], as high concentrations indicate high food availability. Therefore,

we expected predicted moult areas to overlap with high chlorophyll-*a* concentrations. We found this to be true for WSP (Fig 5). For BBSP chlorophyll-*a* concentration did not differ between areas with high and low scaled probability-of-origin values (Fig 4), and for the northern hemisphere species, the high scaled probability-of-origin areas had relatively low chlorophyll-*a* concentrations (Figs 2 and 3). These contrasting findings may be due to large areas being designated as high scaled probability-of-origin areas for the northern species, as the considered possible moulting areas were smaller than those of the southern species, and thus included a smaller range of oceanic $\delta^{13}$C and $\delta^{18}$O values. Indeed, the locations at which the species were observed during the non-breeding period were generally in areas with high chlorophyll-*a* concentrations (Table 5).

For both northern hemisphere species, the predicted moulting areas (Figs 2 and 3) differed considerably from each other, while the predicted WSP moulting areas (Fig 5) were relatively similar. However, the differences in the similarity between scaled probability-of-origin maps may be due to differences in considered possible moulting areas between northern and southern hemisphere species, or an artefact of the scaling procedure. The locations at which the species were observed where generally located within the areas with the 25% highest scaled probability-of-origin values (Table 5). We hypothesised that ESP would moult close to the African West coast, which our predictions of moult locations in the eco-realms also reiterate (Table 6). However, our model also predicted ESP to moult close to Temperate South America (Table 6), where they have not been observed (Fig 2). The other three species were expected to be more widespread, and could be more easily assigned to eco-realms. These findings imply that although predicting moult distribution based on $\delta^{13}$C and $\delta^{18}$O can only be performed at a large geographical scale, and while observation likelihood is highly affected by differences in observation effort, combining both approaches may give an approximate estimate of important moulting areas.

## Context of the study

Stable isotope analyses can only provide large scale movement information, which is one of the study limitations. Stable carbon isotope ($\delta^{13}$C) signatures in seabirds vary depending on phytoplankton distribution [25] and could be a subject of seasonal changes. Phytoplankton distribution varies between years and seasons, and is affected by multiple inorganic processes, such as sea surface temperature, nutrient levels linked to the stratification of the water column (in itself affected by upwelling and turmoil due to waves breaking), $CO_2$ uptake [32], and El Niño events [102]. Due to the striation of water masses around Antarctica, $\delta^{13}$C can be used to predict moulting latitudes in the southern species [76], but only up to the Subtropical Front [50]. Oceanic stable oxygen isotope ratios ($\delta^{18}$O) generally decrease closer to shore in estuarine environments such as the Amazon river mouth and Rio de la Plata area [27], due to a combination of increased freshwater input closer to shore (e.g. river mouths and precipitation) and differences in evaporation rates, creating stark differences with neighbouring marine areas (S1).

Due to these limitations in resolution, while we were able to show inter- and intra-specific differences in moulting distributions, we could only show estimates of predicted moulting areas and assign them to eco-realms close to the shore, while the vast pelagic areas are not included in eco-realms classification [38]. We based our prediction of moulting areas on multi-year isoscapes of stable carbon and oxygen isotopes, instead of isoscapes generated for particular non-breeding periods. While there is a trophic component to the observed $\delta^{13}$C in seabird tissues of approximately 0.8‰ per trophic level [103], this can be controlled for, and therefore $\delta^{13}$C can be used to predict differences in moulting distribution [25]. The trophic

component of $\delta^{18}$O in wild animals is complex and may vary depending on diet and location [104]. We did not know the enrichment factors for $\delta^{18}$O, and could only base those on a very small sample size of regrowing feathers [105]. We could only calculate those enrichment factors between feather and water samples, instead of between feather, different prey items and seawater samples. Calculating enrichment factors between prey and predator, and prey and seawater masses may have provided a deeper understanding of $\delta^{18}$O assimilation [104]. Additionally, while rectrix feathers are often moulted simultaneously with other flight feathers [14, 24], storm-petrels also limit the number of feathers moulted at once [13]. Therefore, while the feathers sampled for our study have grown over a period of several weeks [59], they only represent part of the moulting period.

Nevertheless, our study is the first trying to discover differences in moulting distributions and reconstruct the location of moulting areas of the small storm-petrels breeding in north and south hemispheres based on multiple isotopes. It has filled an evident gap in knowledge about isotopic niches of moulting pelagic storm-petrels. Effective conservation actions and assessments require well-documented knowledge on species' biology and habitat use. While such information is often available for the breeding period, it is frequently lacking for the non-breeding period, especially in pelagic species. Studies such as ours, to better identify important moulting grounds, are therefore needed to properly delineate key conservation areas, and to decide where to direct protection efforts and form conservation planning in the vast marine ecosystem [106]. Combining the large-scale estimates for moulting areas based on $\delta^{13}$C and $\delta^{18}$O with chlorophyll-$a$ concentrations and long-term observation data may provide valuable insights into potential moulting areas. Thus, our study may help to comprehend the year-round feeding ecology of small storm-petrels and understand possible pathways of contaminant (e.g. pollution, microplastics) transfer to breeding areas.

## Conclusions

We found both inter- and intra-specific differences in isotopic moulting ranges for the four studied storm-petrel species. Within ESP, LSP and WSP individuals could be grouped into different moulting niches as the $\delta^{13}$C and $\delta^{18}$O signatures of their tail feathers differed between groups. These divisions were linked to interannual differences in all three species, but also to morphological and $\delta^{15}$N differences in ESP and WSP. These morphological differences were likely caused by differences in foraging ecology and prey availability, rather than sexual segregation. Our results suggest that predicting moult distribution based on $\delta^{13}$C and $\delta^{18}$O can be performed at large geographical scales, but combining these predictions with observational data can be effective to better determine important moulting areas.

Our findings indicate the importance of a large array of different marine regions as moulting areas for the storm-petrel species from both hemispheres because individuals breeding at the same location may adopt different migration strategies, spending the moulting period in different areas. Future studies combining GPS or GLS-tracking and stable isotope analyses based on individuals sampled in multiple locations, including the non-breeding period and multiple feather types, are required to more accurately define moulting areas and further comprehend the foraging ecology at this phase of the annual cycle.

## Supporting information

**S1 File. Base maps for stable oxygen and carbon ocean isoscapes and chlorophyll-$a$ concentrations.**
(DOCX)

**S2 File. European storm-petrel results including two outliers.**
(DOCX)

**S3 File. Data collected by the authors and used in this study.**
(XLSX)

## Acknowledgments

C. Trueman gracefully provided $\delta^{13}$C prediction isoscapes. We would like to thank the Henryk Arctowski Polish Antarctic Station and the Department of Antarctic Biology, the Polish Academy of Sciences for their hospitality and logistic support during our stay on King George Island. We would like to thank Jens-Kjeld Jensen and Marita Gulklett for their hospitality and support for the project on the Faroe Islands. We are grateful to Rosanne Michielsen for her support in the field on King George Island, and Rachel Shepherd, Jessica Hey, Jón Aldará and Katherine Keogan for their support in the field on the Faroe Islands.

## Author Contributions

**Conceptualization:** Anne N. M. A. Ausems, Dariusz Jakubas.

**Data curation:** Anne N. M. A. Ausems.

**Formal analysis:** Anne N. M. A. Ausems, Grzegorz Skrzypek, Katarzyna Wojczulanis-Jakubas.

**Funding acquisition:** Dariusz Jakubas.

**Investigation:** Anne N. M. A. Ausems, Katarzyna Wojczulanis-Jakubas, Dariusz Jakubas.

**Methodology:** Anne N. M. A. Ausems, Katarzyna Wojczulanis-Jakubas, Dariusz Jakubas.

**Project administration:** Anne N. M. A. Ausems, Dariusz Jakubas.

**Resources:** Grzegorz Skrzypek.

**Supervision:** Dariusz Jakubas.

**Validation:** Anne N. M. A. Ausems, Dariusz Jakubas.

**Visualization:** Anne N. M. A. Ausems.

**Writing – original draft:** Anne N. M. A. Ausems.

**Writing – review & editing:** Grzegorz Skrzypek, Katarzyna Wojczulanis-Jakubas, Dariusz Jakubas.

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
