## [Decision Letter · Decision Letter 0]

3 Nov 2020

PONE-D-20-32308

Birds of a feather moult together: differences in moulting range of four species of storm-petrels

PLOS ONE

Dear Dr. Ausems,

Thank you for submitting your manuscript to PLOS ONE. After careful consideration, we feel that it has merit but does not fully meet PLOS ONE’s publication criteria as it currently stands. Therefore, we invite you to submit a revised version of the manuscript that addresses the points raised during the review process.

Please carefully follow recommendations of the three reviewers, specially those related with (1) better contextualising stable isotopes methods used in this field during introduction, like suggested by reviewer 3, (2) more clarification in what novelty the results bring to this field and (3) better compare/ discuss their findings in relation to the broader literature in the topic, like suggested by reviewer 1.

We look forward to receiving your revised manuscript.

Kind regards,

Vitor Hugo Rodrigues Paiva

Academic Editor

PLOS ONE

Journal Requirements:

3. Please upload a copy of Figure 7, to which you refer in your text on page 23, 24, 32 and 33. If the figure is no longer to be included as part of the submission please remove all reference to it within the text.

4.We note that [Figure(s)  2, 3, 4, 5, 6, S1 and S2.2] in your submission contain map/satellite images which may be copyrighted. All PLOS content is published under the Creative Commons Attribution License (CC BY 4.0), which means that the manuscript, images, and Supporting Information files will be freely available online, and any third party is permitted to access, download, copy, distribute, and use these materials in any way, even commercially, with proper attribution. For these reasons, we cannot publish previously copyrighted maps or satellite images created using proprietary data, such as Google software (Google Maps, Street View, and Earth). For more information, see our copyright guidelines: http://journals.plos.org/plosone/s/licenses-and-copyright.

1.    You may seek permission from the original copyright holder of Figure(s) [2, 3, 4, 5, 6, S1 and S2.2] to publish the content specifically under the CC BY 4.0 license. 

5. Please include a caption for figure 7.

Reviewers' comments:

Reviewer's Responses to Questions

**Comments to the Author**

1. Is the manuscript technically sound, and do the data support the conclusions?

Reviewer #1: Partly

Reviewer #2: Yes

Reviewer #3: Yes

2. Has the statistical analysis been performed appropriately and rigorously? 

Reviewer #1: Yes

Reviewer #2: No

Reviewer #3: I Don't Know

3. Have the authors made all data underlying the findings in their manuscript fully available?

Reviewer #1: Yes

Reviewer #2: Yes

Reviewer #3: Yes

4. Is the manuscript presented in an intelligible fashion and written in standard English?

Reviewer #1: No

Reviewer #2: Yes

Reviewer #3: Yes

5. Review Comments to the Author

Reviewer #1: In the manuscript “Birds of a feather moult together: differences in moulting range of four species of storm-petrels”, Ausmes and colleagues aim was to determine the moutling locations of 2 northern and 2 southern species of storm-petrels using carbon and oxygen stable isotopes from feather and oceanic isoscapes. While I realized that the aim of the manuscript is quite daunting, the results are hard to interpret and make for a weak conclusion. Beside the methods used in this manuscript, there is little novelty we learn regarding the moulting regions of those 4 species that is not already in the species accounts. Furthermore, the discussion is based on the moulting area of just one feather type and surely, the moulting period is a lot longer than that and could span over a much wider region.

The paper would greatly benefit from reformatting, especially in the results section, where a lot of text could be easily summarized in a few tables and make the results easier to read (and therefore easier to understand). In addition, the grammar could be improved throughout.

Specific comments

Abstract

Introduction

L50. Need a reference. Maybe Fayet et al. 2016. https://doi.org/10.1111/1365-2656.12580

L61. … “more time spend on the water than outside of moulting”. Awkward, suggest rephrasing.

L64. This statement, to which I agree, is somewhat contradictory to your methods. Let me explain: you are saying that the moult process extend over a large part of the non-breeding period, yet you only look at one type of feather for your analysis (the outer rectrices). It might have been interesting to also look at other feather type (flight feathers for example).

L69. … “and prematurely failed to track”. Awkward, suggest rephrasing.

L72. 495km. Yes, it is a low accuracy, but the paper you cite indicate that this accuracy can be lowered to ~ 50 km with the appropriate analysis. Furthermore, what is 500 km in the vast of the Pacific or Atlantic Oceans compared to the results obtained from stable isotopes?

L84. Maybe give the readers an idea of the resolution scale obtained this way.

L90-91. I agree that WSP are considered the world’s most abundant seabird species, but this sentence needs to be rephrased.

L96-97. How do you justify that knowledge about the moulting period is more important than the rest of the non-breeding season? Expand a bit on this topic.

L99. Maybe specify here that you used just one feather type.

L107. Add a coma after size-dimorphic species.

Materials and methods

L164. How can you be sure that by sampling individuals caught in mist-nets, you have only adults, and not also pre-breeders? What do we know about the moult schedule of pre-breeders and/or non-breeders?

L164. It would be very good to have the number of each species caught over the years.

L170. Not everyone is familiar with ptilochronology, and it would be good to give a short description of the methods, in addition to the reference.

L172-185. Briefly explain why not all birds were sexed.

L206-207. There are small squares after the references.

L230. I am surprised you included sex in the explanatory variables for the CIT analysis, considering that you did not have the sex of some (for some species) or most (for other species)? Please explain.

L293-295. I am concern by the low sample size to calculate the discrimination factors between 18O of ocean and feathers (n=5 for northern species, n=3 for southern species).

Results

L344-352. This entire paragraph could be nicely summarized in a small table, with all relevant information. It would be a lot easier to read and understand. In addition, it would be good to add a sample size column in that table.

L353. Apparently, those outliers were already removed from the values given above (ESP δ13C values ranged between -20.7 and -16.9).

L397. Not all individuals were sexed, so why not used only wing length?

L358-427. This entire section is rather thick to read and would probably be better in a table. Table 1 could be the starting point, adding the statistical results in extra columns. It would also somewhat shorten the results section.

Discussion

L585-587. As mentioned in the general comments, I think the authors should be careful in the conclusion of the moutling area of those 4 storm-petrels species based on just one feather. They have to keep in mind, that the moulting period will extend over a wider area, and write their discussion accordingly.

L587-599. This is a repeat of the results and not the discussion.

L627. Add a space after values.

L637-640. This is a repeat of the results. The discussion should discuss the results, not repeat them. For example, given the information you give L645-646, you may be able to conclude that the difference in ESP moulting area is not sexual segregation.

Figure 1. The sum of the numbers after each node does not add up to the number of individuals of each species in the supplementary spreadsheet.

Figures 2-5. Please include latitude longitude axis on the maps. The sentence “The 95 % quantile of the scaled probability-of-origin values per terminal CIT node are shown with the dashed line” should come after the description of the first part of the figures. The legends should indicate that the white dots show observations locations retrieved from online database.

Figure 5. The maps are small, consider putting them on two different rows for better visibility.

Figure 6. The choice of color makes it hard to read the map and is not suitable for people with color-vision deficiency.

Reviewer #2: This study by Ausems et al. is the first study trying to discover differences in moulting distributions and to model such areas for storm-petrels. I think it is a study of interest since most wintering areas and important information on the ecology of storm-petrels is still unknown. However, there are some considerations that the authors should take and review before publishing.

Major comments

L283-289 - Firstly, when correcting the differences of feather isotopic values and source material, the authors use a use 0.8 ‰ enrichment factor per trophic level, and assume that all study species consume zooplankton. However, there are studies that has showed that some storm-petrels might feed on prey of higher trophic levels, and inclusively Hedd and Montevecchi – a study you reference - showed that LSP might eat mesopelagic fish (Myctophidae and Gadidae), crustaceans or even cephalopods. Although I understand that it is what is mostly described for the majority of storm-petrels, the authors should consider this and replace “they consume zooplankton” (in line 287) by “they consume mostly zooplankton”. Also, I think that authors should consider more references when using a fixed value for enrichment factor: most literature present higher values (around 1 ‰), and so there is not a fixed number for the enrichment of C13 per trophic level in seabirds, nor it has been studied for procellariiforms. A study by Meier et al (2017) has calculated a mean diet-feather trophic enrichment to apply on Balearic shearwater, based on different literature, and they obtained 1,9 +/- 0,5 ‰ for C13. To be more accurate, I think you should consider calculate an average value based on the literature of seabirds that have a similar ecology as storm-petrels, and recalculate to check if there is any significant differences.

Meier, R. E., Votier, S. C., Wynn, R. B., Guilford, T., McMinn Grivé, M., Rodríguez, A., Newton, J., Maurice, L., Chouvelon, T., Dessier, A., & Trueman, C. N. (2017). Tracking, feather moult and stable isotopes reveal foraging behaviour of a critically endangered seabird during the non-breeding season. Diversity and Distributions. https://doi.org/10.1111/ddi.12509

L330 – You mention that there are differences between years in the observation efforts, and so this observational data must be interpreted with care. Other thing to take into consideration is that the observations recorded in the repositories are not only from your breeding colonies. For example, Pollet et al. (2014) has showed that colonies of LSP that are only 380 km apart can have different wintering distributions. You should consider this and add this point in the discussion.

Pollet, I.L., Hedd, A., Taylor, P.D., Montevecchi, W.A. & Shutler, D. (2014). Migratory movements and wintering areas of Leach’s Storm-Petrels tracked using geolocators. J. F. Ornithol., 85, 321–328

L660-662 - I think you should tone down the certainty that your differences are not caused by sexual segregation. Although you did not found significant effect of sex in CIT, you have some limitations, specially the fact that in ESP sex has a high correlation with wing length in the models. I would suggest consider this, and replace the conclusion by “ (…) different moulting distributions are probably not caused by sexual segregation (…)”.

Minor comments:

L24 – Place a space between “seabird” and “is”.

L28 – You use Oceanodroma leucorhoa, however, this nomenclature has been replaced by BirdLife International as Hydrobates leucorhous, and recent studies have been adopting it. Please consider replace it along the text.

L71 – I think replacing “method to study year-round movements and the position (..)” by “method to study year-round movements, as well as the position (…)” makes the sentence clearer.

L76 – Replace “thus represent the stable isotope composition of the prey eaten during feather synthesis” by “thus represent the signatures of the prey eaten during feather synthesis” to avoid repetition.

L102 – It is not clear what source of N15 are you referring to. Further in the methods you explain that it is from the feathers, but would be easier to clarify here straight away. It is not clear in the abstract either, so the reader keeps not understanding from where this comes from.

L172-174 - Is not very clear if both feathers of WSP/BBSP and blood of ESP/LSP were used for molecular sexing. If it is so, maybe clarify it by writing “For molecular sexing, we collected several body feathers from the back of the neck from each individual of WSP and BBSP, and a drop of blood from ESP and LSP individuals, stored in 70 % ethanol”.

L177-178 - If you changed the annealing temperature, you should state that is an adaptation of the protocol by Griffiths et al. 1998. Also replace “The primers amplify (…) by “The primer pair amplify (…)” in line 178.

L345-353 - Most of these values are already in the table 1, and so this becomes repetitive and difficult to digest. I would mention only in which species where the highest and lowest values of each isotope was detected, and mention the table 1 right here.

L363 – There is not any p values nor other t-test values represented in table 1. Delete this here.

L372 / L586 / L705-706 – I am almost sure that the words “stable isotopes” comes always together, so please check it and replace by “nitrogen stable isotope ratio”, “Carbon and oxygen stable isotopes” and “oxygen stable isotope ratio”.

L433-436 - Maybe it would be beneficial to include in the methods the thresholds for when is considered an overlap in similarity/similar (over 0,6 if not mistaken). Also, in L434, add “than ESP” in the sentence “For LSP the similarity was higher than ESP, but still (…)” to make it clearer.

L411 - Please be consistent in decimal presentation throughout the text, you have “δ15N values ≤ 14.787” when you have presented values with only two decimals until then.

Table 4 – Not sure if it was a formatting problem, but again try to be consistent in the number of decimals you present. For example you have -0.5 for Temperate Northern Atlantic and 0.8 in Temperate Southern Africa, when it should be -0.50 and 0.80 respectively.

L597-600 - I don’t understand how “they also generally have a more pelagic lifestyle (35,43) with larger foraging areas where stable isotope ratios differ over larger areas” explains the lack of division into subgroups. Maybe cut this phrase into two, and clarify the second part better.

L614-616 – This sentence is a bit confusing to read. I would suggest replacing by “Indeed, year was only a significant factor in the WSP groups predicted to have moulting ranges extending further South (Table 2; Figure 1B and 5).”

L629 – Add a space between “values” and “the”.

L678-681 - This sentence is too long, please add a comma after “origin areas for the northern species, as the (…)” or cut it in two sentences.

L751 – Add a comma and “the” in the sentence “migration strategies, spending the moulting”.

Reviewer #3: This study by Ausems et al. measures stable isotope ratios of carbon, nitrogen and oxygen in feathers sampled from adults of multiple storm petrel species (European; Leach’s; black-bellied; Wilson’s) in the Northern and Southern hemispheres. Feathers are metabolically inert after synthesis and hence their isotopic composition reflects diet during synthesis. This is a well-established method in seabird ecology. The authors examined factors driving variation in feather δ13C and δ18O values and made predictions about moulting locations by using isoscapes. Overall the study is well designed and comprehensive; however, there are some issues that I believe the authors should address. The Introduction would benefit from a clearer explanation of the use and interpretation of stable isotope ratios. How, for instance, should ecologists interpret stable isotope ratios of oxygen? These are not as common in the seabird literature as those of carbon and nitrogen. Moreover, references to the isotopic niche in the Introduction are confusing given the lack of emphasis placed on δ15N values, which reflect an important niche axis. Please clarify and consider referencing some important literature on the isotopic niche (e.g. Newsome et al. 2007). The Introduction may also benefit from some more specific information on the diets and foraging ecology of storm petrels. Regarding the Materials and methods, the stable isotope analysis section requires greater precision; for instance, where is the explanation of the delta notation? The authors should also be careful not to refer to δ13C and δ18O values as elements. I am not familiar with the conditional inference tree technique, but do the authors consider interactions among the various predictor variables of feather δ13C and δ18O values? The authors may consider a separate heading for the ethics statement. The opening paragraph of the Results is quite dense, and in my opinion would be better suited to a table. The authors might wish to consider sub-headings to organise the Discussion.

6. PLOS authors have the option to publish the peer review history of their article (what does this mean?). If published, this will include your full peer review and any attached files.

Reviewer #1: No

Reviewer #2: **Yes: **Ana Rita Carreiro

Reviewer #3: No

---

## [Author Response · Author response to Decision Letter 0]

8 Dec 2020

Editor comments

(1) better contextualising stable isotopes methods used in this field during introduction, like suggested by reviewer 3, 

Response: We added more information about δ13C and δ18O interpretation.

Introduction, page 5, lines 106 – 109: Stable isotope compositions of both elements vary spatially in marine ecosystems; δ13C values are correlated with phytoplankton distribution [25,26] while δ18O values are correlated with salinity and fresh water input [27]. Therefore, they follow inshore/offshore gradients [26–28].

(2) more clarification in what novelty the results bring to this field and 

Response: We added a line to the introduction explaining the novelty of our study, but see also the response to reviewer #1.

Introduction, page 5, lines 109 – 111: To our knowledge, this study is the first to combine δ13C and δ18O analyses to determine differences in moulting distributions of storm-petrels breeding sympatrically in both hemispheres.

Additionally, we would like to point out that reviewer #2 wrote “I think it is a study of interest since most wintering areas and important information on the ecology of storm-petrels is still unknown,” and reviewer #3 commented “How, for instance, should ecologists interpret stable isotope ratios of oxygen? These are not as common in the seabird literature as those of carbon and nitrogen”. From these responses we inferred that reviewer # 2 and #3 did not question the novelty of the approach, although they did require further explanations.

(3) better compare/ discuss their findings in relation to the broader literature in the topic, like suggested by reviewer 1.

Response: We have added additional references where relevant, as suggested by reviewer 1 and 2, and expanded the discussion. Please see our responses to these reviewers for more details.

Journal requirements

Response: We have changed the formatting and style to meet the style requirements. We have removed the short title from the title page. 

2. Please include captions for your Supporting Information files at the end of your manuscript , and update any in-text citations to match accordingly. Please see our Supporting Information guidelines for more information: http://journals.plos.org/plosone/s/supporting-information.

Response: We have included captions after the reference list.

3. Please upload a copy of Figure 7 , to which you refer in your text on page 23, 24, 32 and 33. If the figure is no longer to be included as part of the submission please remove all reference to it within the text.

Response: This was a formatting error and should have read figure 6, however in accordance with point 4 (below) we have removed this figure.

4.We note that [Figure(s) 2, 3, 4, 5, 6, S1 and S2.2] in your submission contain map/satellite images which may be copyrighted. All PLOS content is published under the Creative Commons Attribution License (CC BY 4.0), which means that the manuscript, images, and Supporting Information files will be freely available online, and any third party is permitted to access, download, copy, distribute, and use these materials in any way, even commercially, with proper attribution. For these reasons, we cannot publish previously copyrighted maps or satellite images created using proprietary data, such as Google software (Google Maps, Street View, and Earth). For more information, see our copyright guidelines: http://journals.plos.org/plosone/s/licenses-and-copyright. […].

1. We note that the data from Spalding et al 2007 are, as noted on https://databasin.org/datasets/3b6b12e7bcca419990c9081c0af254a2, licensed under a CC BY license that doesn't permit commercial use. As the version of the CC BY license that PLOS ONE uses does permit for commercial use, we ask that you request permission from the copyright holder to publish the data in your figure under CC BY 4.0.

Response: Figs 2 – 5 and S2.2 were generated by the prediction models described in the text and are not satellite images. Fig 6 has been removed. The maps in S1 are based on data downloaded from NASA websites or supplied by other researchers. They are referenced in the text and links to the data are provided as well. These data can be freely downloaded and used.

5. Please include a caption for figure 7.

Response: This was a formatting error and should have read figure 6, however in accordance with point 4 this figure has been removed.

Reviewer comments

Reviewer #1

In the manuscript “Birds of a feather moult together: differences in moulting range of four species of storm-petrels”, Ausmes and colleagues aim was to determine the moutling locations of 2 northern and 2 southern species of storm-petrels using carbon and oxygen stable isotopes from feather and oceanic isoscapes. While I realized that the aim of the manuscript is quite daunting, the results are hard to interpret and make for a weak conclusion. 

Response: Thank you for your feedback. We understand that not everyone will be convinced by our methods and conclusions, but reviewer #2 and #3 do seem positive about the research we presented. Stable isotope analysis of feathers is widely used in pelagic seabirds to infer their non-breeding or moulting areas (Cherel et al., 2000; Hobson, 2011). Our aim was to find which factors affect differences in moulting location, and we believe we have done so. Additionally, we tried to predict moulting ranges based on the stable isotope values, but these predictions could only be made on a large scale and we realise that they are not as direct as GLS or GPS tracking data.

Beside the methods used in this manuscript, there is little novelty we learn regarding the moulting regions of those 4 species that is not already in the species accounts. 

Response: We have added a line in the introduction stating the relevance and novelty of our study. The species accounts as described by the IUCN and BirdLife show a very wide, and uncertain range of possible non-breeding areas. While our study also shows they may migrate over large areas, it additionally provides further insight into possible differences in moulting distribution within species and what may influence these. Understanding the mechanisms affecting moulting distribution may not only provide insight into the moulting locations, but it also helps our understanding of what may cause differences in foraging and movement ecology. This is the first study on such a scale for the studied species. 

Introduction, page 5, lines 109 – 111: To our knowledge, this study is the first to combine δ13C and δ18O analyses to determine differences in moulting distributions of storm-petrels breeding sympatrically in both hemispheres.

Furthermore, the discussion is based on the moulting area of just one feather type and surely, the moulting period is a lot longer than that and could span over a much wider region.

Response: As we stated in the manuscript, moulting the rectrix feather takes several weeks. Of course, different feathers may be moulting in different areas, but for ethical reasons we decided not to sample more feathers as that may have negatively affected their flight efficiency (which is especially important for pelagic seabirds), survival and breeding success.

The paper would greatly benefit from reformatting, especially in the results section, where a lot of text could be easily summarized in a few tables and make the results easier to read (and therefore easier to understand). In addition, the grammar could be improved throughout.

Response: We added several more tables (i.e. Tables 1 and 3) to reduce the density of statistical results provided in the text. We have corrected the instances you identified where we made grammatical or style errors.

L50. Need a reference. Maybe Fayet et al. 2016. https://doi.org/10.1111/1365-2656.12580

Response: Fayet et al. 2016 experimentally prolonged the breeding period and tested the effects of this treatment on the subsequent breeding period. As such, this is not a good reference for our manuscript as we are interested in carry-over effects of events from the non-breeding period. Instead, we changed the order of the next sentence in which we elaborated on the possible carry-over effects from the non-breeding period into the subsequent breeding season. 

Introduction, page 2, lines 52 – 54: Differences in non-breeding distribution and in food availability [1], diet quality [2] or diet composition [3] at the non-breeding area may affect survival [4] and breeding success [5] during the subsequent breeding season.

L61. … “more time spend on the water than outside of moulting”. Awkward, suggest rephrasing.

Response: We have changed this sentence.

Introduction, page 3, lines 63 – 66: In pelagic seabirds moulting individuals may spend more time floating on the water than outside of the moulting period [12], affecting foraging effectiveness.

L64. This statement, to which I agree, is somewhat contradictory to your methods. Let me explain: you are saying that the moult process extend over a large part of the non-breeding period, yet you only look at one type of feather for your analysis (the outer rectrices). It might have been interesting to also look at other feather type (flight feathers for example).

Response: While you are correct that we only considered one feather type, which does not constitute the entire moulting period, here we describe moult in general. We did not yet zoom in on our specific research aims, but were rather putting our study into perspective. Additionally, while we agree with the sentiment of the reviewer, we do not believe the second paragraph of the introduction is the right place to address potential limitations of our study. Rather, we addressed these concerns in the discussion (see Discussion; Context of the study, pages 40 – 42, lines 801 – 831).

L69. … “and prematurely failed to track”. Awkward, suggest rephrasing.

Response: We have changed this sentence. 

Introduction, page 4, lines 74 – 75: […] and incomplete tracks [15–17].

L72. 495km. Yes, it is a low accuracy, but the paper you cite indicate that this accuracy can be lowered to ~ 50 km with the appropriate analysis. Furthermore, what is 500 km in the vast of the Pacific or Atlantic Oceans compared to the results obtained from stable isotopes?

Response: We agree with this statement, and we have removed the reference to the position estimation.

Introduction, page 4, lines 75 – 77: Additionally, geolocators, while not proven detrimental [15–17], may be considered a relatively invasive method to study year-round movements.

L84. Maybe give the readers an idea of the resolution scale obtained this way.

Response: We cannot put any specific number on the geographic resolution of the method we use, but we have given a relative scale. 

Introduction, page 4, lines 88 – 89: However, combining multiple isotopes may considerably increase resolution to a more regional scale.

L90-91. I agree that WSP are considered the world’s most abundant seabird species, but this sentence needs to be rephrased.

Response: We have rephrased this sentence to make it easier to read.

Introduction, page 5, lines 94 – 97: The storm-petrels, and in particular WSP, are considered the world’s most abundant seabird species. However, relatively little is known about their ecology during the non-breeding period.

L96-97. How do you justify that knowledge about the moulting period is more important than the rest of the non-breeding season? Expand a bit on this topic.

Response: We have removed the reference to the moulting period. We felt like we would repeat ourselves if we expand on its importance as it was already discussed in the second paragraph of the introduction.

Introduction, page 5, lines 100 – 102: As such, they could be used as valuable sentinel species [23], but for that more knowledge is needed about their ecological niches during the non-breeding period.

L99. Maybe specify here that you used just one feather type.

Response: We included which feather type we studied. 

Introduction, page 5, line 104: […] we used the stable isotope composition of two elements: δ13C and δ18O of tail feathers […]

L164. How can you be sure that by sampling individuals caught in mist-nets, you have only adults, and not also pre-breeders? What do we know about the moult schedule of pre-breeders and/or non-breeders?

Response: We do not, and we do not claim to do so. We are interested in all possible moulting locations, and not only for breeders. There are differences in moult schedules between pre-breeders and non-breeders in European storm-petrels (Arroyo et al., 2004) but this does not affect the reconstruction of the moulting distribution.

L164. It would be very good to have the number of each species caught over the years.

Response: We have included those numbers to the study species section of the materials and methods. Additionally, we have included sample sizes analysed per species to Table 1.

Materials and methods; Study species, page 7, lines 140 – 144: We studied ESP and LSP in August of 2018 (n = 52; n = 56, respectively) and 2019 (n = 40; n = 37, respectively) on the island of Mykines, Faroe Islands (62°05´N, 07°39´W), and BBSP and WSP during the austral summer of 2017 (n = 15; n = 100, respectively) and 2018 (n = 19; n = 126, respectively) around the Henryk Arctowski Polish Antarctic Station, on King George Island, South Shetland Islands, Antarctica (62°09´S, 58°27´W).

L170. Not everyone is familiar with ptilochronology, and it would be good to give a short description of the methods, in addition to the reference.

Response: We have included a short explanation of ptilochronology.

Materials and methods; Data collection; Field study, page 9, lines 192 - 195: We determined the feather growth rate for the outermost rectrix by measuring growth bar width to the nearest 0.1 mm × d-1. Feather growth bars are visible as alternating light and dark bands, formed during feather synthesis, but see Ausems et al. 2019 [59] for a detailed description of the method used.

L172-185. Briefly explain why not all birds were sexed.

Response: A subset of samples was selected for sexing, but some of the samples did not give reliable PCR products. Since the analyses performed on the subset with successfully sexed birds did not reveal any significant effect, we did not continue the lab work. We have included a short explanation to the manuscript. 

Materials and methods, Data collection; Molecular sexing; page 9, lines 207: Some of the samples did not give reliable PCR products, […].

L206-207. There are small squares after the references.

Response: This seems to be a formatting issue happening during the creation of the review pdf. We have tried to fix it on our end.

L230. I am surprised you included sex in the explanatory variables for the CIT analysis, considering that you did not have the sex of some (for some species) or most (for other species)? Please explain.

Response: Since we did have sex for a part of the sample, we decided to use the data available, instead of discarding everything because some data was missing. Even though we tested only a subset the CIT method used should be able to show whether sex was a variable of importance. Additionally, reviewer #2 did seem to see the relevance of using the sex data we collected.

L293-295. I am concern by the low sample size to calculate the discrimination factors between 18O of ocean and feathers (n=5 for northern species, n=3 for southern species).

Response: We are too, but ecological field studies very rarely provide perfect data sets, especially when they are difficult to obtain and originate from remote locations. At the moment, there are no better discrimination factors available. We address this limitation in the discussion (see Discussion; Context of the study, pages 40 – 42, lines 802 – 832).

L344-352. This entire paragraph could be nicely summarized in a small table, with all relevant information. It would be a lot easier to read and understand. In addition, it would be good to add a sample size column in that table.

Response: We have added an extra table (Table 1) containing the statistical results and a sample size column. 

L353. Apparently, those outliers were already removed from the values given above (ESP δ13C values ranged between -20.7 and -16.9).

Response: Yes, the outliers were removed. We moved this statement to the materials and methods section. 

Materials and methods; Statistical analyses, page 11, lines 240 – 244: From ESP two apparent outliers with δ13C < -23 ‰ were removed for further analyses, as we could not determine whether these values were caused by biological processes (i.e. different moulting ranges or ages) or were due to measurement errors. The CIT and probability-of-origin results for the analyses including the outliers are reported in the supplementary data (S2).

L397. Not all individuals were sexed, so why not used only wing length?

Response: Because while wing length may be associated with sex, sex is not the only parameter affecting wing length and wing length may have a separate effect on moulting distribution. But see also our response to the comment at L230. 

L358-427. This entire section is rather thick to read and would probably be better in a table. Table 1 could be the starting point, adding the statistical results in extra columns. It would also somewhat shorten the results section.

Response: We have created two extra tables (Tables 1 and 3) to hold statistical results, though not all statistical results were removed from the results section.

L585-587. As mentioned in the general comments, I think the authors should be careful in the conclusion of the moutling area of those 4 storm-petrels species based on just one feather. They have to keep in mind, that the moulting period will extend over a wider area, and write their discussion accordingly.

Response: We have more clearly stated that we analysed rectrix moult. However, as rectrix moult takes several weeks, we believe our results represent at least an important portion of the moulting period.

Discussion, page 36, line 686: […] the storm-petrel species moulted their rectrices in different areas […].

L587-599. This is a repeat of the results and not the discussion.

Response: We have removed part of this section, but not all as we also interpret the results. 

Discussion, page 35, lines 687 – 698: ESP had significantly higher δ13C values and lower δ18O values than LSP, suggesting less pelagic moulting ranges [93], while BBSP had both higher δ13C and higher δ18O values than WSP indicating a more pelagic lifestyle and moulting grounds further North [76,93]. The lack of any division into subgroups in BBSP is likely due to the relatively small sample size (n = 19).

L627. Add a space after values.

Response: We have changed this sentence accordingly.

L637-640. This is a repeat of the results. The discussion should discuss the results, not repeat them. For example, given the information you give L645-646, you may be able to conclude that the difference in ESP moulting area is not sexual segregation.

Response: While this is true, we included this information here to show what results we are referring to. Therefore, we decided to leave this sentence unchanged.

Figure 1. The sum of the numbers after each node does not add up to the number of individuals of each species in the supplementary spreadsheet.

Response: This is due to missing data. We have removed the individuals that did missed either δ13C or δ18O values from the analyses, but did use them to determine the relationship between wing length and sex. We included a line explaining this at the beginning of the statistical analyses section in the materials and methods. 

Materials and methods; Statistical analyses, page 11, lines 239 – 240: Individuals missing δ13C or δ18O values were removed from further analyses.

Figures 2-5. Please include latitude longitude axis on the maps. The sentence “The 95 % quantile of the scaled probability-of-origin values per terminal CIT node are shown with the dashed line” should come after the description of the first part of the figures. The legends should indicate that the white dots show observations locations retrieved from online database.

Response: We have included latitude/longitude axis titles; however, we do not see the point of adding a legend only for the white dots. Their meaning is explained in the caption as well as the meaning of the other symbols. Moving the mentioned sentence would break up the panel description the way we interpreted this comment, so we preferred to keep it in the current order. 

Figure 5. The maps are small, consider putting them on two different rows for better visibility.

Response: We have changed the layout of the figure.

Figure 6. The choice of color makes it hard to read the map and is not suitable for people with color-vision deficiency.

Response: In accordance with the requirements of the journal, we have removed this figure altogether.

Reviewer #2

This study by Ausems et al. is the first study trying to discover differences in moulting distributions and to model such areas for storm-petrels. I think it is a study of interest since most wintering areas and important information on the ecology of storm-petrels is still unknown. However, there are some considerations that the authors should take and review before publishing.

Response: Thank you for the valuable feedback. We are glad you think this study is interesting, and we considered all comments carefully.

L283-289 - Firstly, when correcting the differences of feather isotopic values and source material, the authors use a use 0.8 ‰ enrichment factor per trophic level, and assume that all study species consume zooplankton. However, there are studies that has showed that some storm-petrels might feed on prey of higher trophic levels, and inclusively Hedd and Montevecchi – a study you reference - showed that LSP might eat mesopelagic fish (Myctophidae and Gadidae), crustaceans or even cephalopods. Although I understand that it is what is mostly described for the majority of storm-petrels, the authors should consider this and replace “they consume zooplankton” (in line 287) by “they consume mostly zooplankton”.

Response: We have changed the sentence accordingly, and included a new reference (Frith et al., 2020) that shows that LSP diets contain increasing amounts of crustaceans. We do know that the storm-petrels may forage on a wider range of prey, but for the non-breeding period not enough data is available to make more detailed decisions. However, see also our response to the comment below.

Materials and methods; Statistical analyses; Predicted moulting areas, page 14, lines 321 – 322: Although the exact trophic level of the studied storm-petrels during the non-breeding period is unknown, they consume mostly zooplankton [53,77-87] […].

Also, I think that authors should consider more references when using a fixed value for enrichment factor: most literature present higher values (around 1 ‰), and so there is not a fixed number for the enrichment of C13 per trophic level in seabirds, nor it has been studied for procellariiforms. A study by Meier et al (2017) has calculated a mean diet-feather trophic enrichment to apply on Balearic shearwater, based on different literature, and they obtained 1,9 +/- 0,5 ‰ for C13. To be more accurate, I think you should consider calculate an average value based on the literature of seabirds that have a similar ecology as storm-petrels, and recalculate to check if there is any significant differences.

Meier, R. E., Votier, S. C., Wynn, R. B., Guilford, T., McMinn Grivé, M., Rodríguez, A., Newton, J., Maurice, L., Chouvelon, T., Dessier, A., & Trueman, C. N. (2017). Tracking, feather moult and stable isotopes reveal foraging behaviour of a critically endangered seabird during the non-breeding season. Diversity and Distributions. https://doi.org/10.1111/ddi.12509

Response: While we understand the concerns raised here, we would like to point out that an average value based on literature would likely be close to 0.8 ‰. However, what may be more convincing is the fact that using a different correction method for δ13C does not change the results of the isocat prediction markedly. Below, we have included two maps showing such: (A) shows the results for ESP when using the 0.8 ‰ trophic enrichment factor, and (B) shows the results for ESP when using a linear regression model (Bearhop et al., 2006; Ceia et al., 2014, 2012). In this model we regressed δ13C and δ15N, to control for the trophic component in δ13C. We then corrected the observed δ13C values by subtracting the residuals of the linear regression model.

Therefore, we have taken the advice of the reviewer to include a wider range of references, but we decided not to change the analyses we performed.

Materials and methods; Statistical analyses; Predicted moulting areas, page 14, lines 318 – 321: For seabirds, δ13C increases with trophic level, with trophic enrichment factors varying between species, sampled tissues and diet [74]. In storm-petrels, a trophic enrichment factor of 0.8 ‰ per trophic level has been used in previous studies [75,76].

L330 – You mention that there are differences between years in the observation efforts, and so this observational data must be interpreted with care. Other thing to take into consideration is that the observations recorded in the repositories are not only from your breeding colonies. For example, Pollet et al. (2014) has showed that colonies of LSP that are only 380 km apart can have different wintering distributions. You should consider this and add this point in the discussion.

Pollet, I.L., Hedd, A., Taylor, P.D., Montevecchi, W.A. & Shutler, D. (2014). Migratory movements and wintering areas of Leach’s Storm-Petrels tracked using geolocators. J. F. Ornithol., 85, 321–328

Response: Your assumptions about differences between breeding colonies are correct, however we studied only one colony at a time. We did not interpret the observational data as proof that individuals from the studied colonies were seen in specific areas, but rather to verify the probability of actual utilization of the predicted moulting areas by the studied species. If certain areas did not contain any observations of the species, but had high probability-of-origin values, we would have assumed that the probability-of-origin values did not reflect the likelihood of the species using this moulting ground.

L660-662 - I think you should tone down the certainty that your differences are not caused by sexual segregation. Although you did not found significant effect of sex in CIT, you have some limitations, specially the fact that in ESP sex has a high correlation with wing length in the models. I would suggest consider this, and replace the conclusion by “ (…) different moulting distributions are probably not caused by sexual segregation (…)”.

Response: We have changed the sentence accordingly. However, neither sex nor wing length had a significant effect on the CIT results for ESP.

Discussion, page 39, lines 761 – 763: Our results thus suggest that the different moulting distributions are probably not caused by sexual segregation in ESP, LSP or WSP. 

L24 – Place a space between “seabird” and “is”.

Response: We have corrected this sentence.

L28 – You use Oceanodroma leucorhoa, however, this nomenclature has been replaced by BirdLife International as Hydrobates leucorhous, and recent studies have been adopting it. Please consider replace it along the text.

Response: We were aware of the change in nomenclature but decided to use the old name as most papers still do. However, we have included a sentence mentioning the new name in the methods section.

Materials and methods; Study species, page 7, lines 147 – 149: The species name of LSP has therefore recently been changed by BirdLife to Hydrobates leucorhous [43] though the old nomenclature is still widely used as well.

L71 – I think replacing “method to study year-round movements and the position (..)” by “method to study year-round movements, as well as the position (…)” makes the sentence clearer.

Response: In response to reviewer #1 we have removed the second part of the sentence altogether.

L76 – Replace “thus represent the stable isotope composition of the prey eaten during feather synthesis” by “thus represent the signatures of the prey eaten during feather synthesis” to avoid repetition.

Response: We have changed the sentence accordingly.

L102 – It is not clear what source of N15 are you referring to. Further in the methods you explain that it is from the feathers, but would be easier to clarify here straight away. It is not clear in the abstract either, so the reader keeps not understanding from where this comes from.

Response: We have added the required information.

L172-174 - Is not very clear if both feathers of WSP/BBSP and blood of ESP/LSP were used for molecular sexing. If it is so, maybe clarify it by writing “For molecular sexing, we collected several body feathers from the back of the neck from each individual of WSP and BBSP, and a drop of blood from ESP and LSP individuals, stored in 70 % ethanol”.

Response: We have changed this sentence accordingly.

L177-178 - If you changed the annealing temperature, you should state that is an adaptation of the protocol by Griffiths et al. 1998. Also replace “The primers amplify (…) by “The primer pair amplify (…)” in line 178.

Response: We have changed this section.

Materials and methods; Data collection; Molecular sexing; page 9, lines 201 – 204: We followed Griffiths et al. 1998 [60] to perform molecular sexing with primer pair 2550F and 2718R but adapted the protocol by using 50 °C for the annealing temperature in the polymerase chain reaction. The primer pair amplifies introns […].

L345-353 - Most of these values are already in the table 1, and so this becomes repetitive and difficult to digest. I would mention only in which species where the highest and lowest values of each isotope was detected, and mention the table 1 right here.

Response: We have added a new table (Table 1) including these data. 

L363 – There is not any p values nor other t-test values represented in table 1. Delete this here.

Response: We were referring to the stable isotope values, not the statistical results.

L372 / L586 / L705-706 – I am almost sure that the words “stable isotopes” comes always together, so please check it and replace by “nitrogen stable isotope ratio”, “Carbon and oxygen stable isotopes” and “oxygen stable isotope ratio”.

Response: We have checked this again, but the recommended terms are correct in our manuscript. Please see Coplen 2011 for the terms recommended by The Commission on Isotopic Abundances and Atomic Weigths/IUPAC (Coplen, 2011).

L433-436 - Maybe it would be beneficial to include in the methods the thresholds for when is considered an overlap in similarity/similar (over 0,6 if not mistaken). 

Response: The Jaccard index is expressed as the proportion of overlap, and the threshold for when two objects are considered similar depends on the nature of the object and the data studied. For example, in our study, the extent of the map area considered for the prediction model affected the similarity between two predictions. For the Northern species this extent was considerably smaller than for the Southern species. Relatively large areas of the maps in WSP had low probability-of-origin values, thus increasing the proportion of overlap, though these results would have been different if we had only considered the Atlantic Ocean. Therefore, we decided not to provide any threshold values.

Also, in L434, add “than ESP” in the sentence “For LSP the similarity was higher than ESP, but still (…)” to make it clearer.

Response: We have changed the sentence accordingly.

L411 - Please be consistent in decimal presentation throughout the text, you have “δ15N values ≤ 14.787” when you have presented values with only two decimals until then.

Response: This was a formatting error and has been corrected accordingly.

Table 4 – Not sure if it was a formatting problem, but again try to be consistent in the number of decimals you present. For example you have -0.5 for Temperate Northern Atlantic and 0.8 in Temperate Southern Africa, when it should be -0.50 and 0.80 respectively.

Response: These were indeed formatting issues, and we have corrected them.

L597-600 - I don’t understand how “they also generally have a more pelagic lifestyle (35,43) with larger foraging areas where stable isotope ratios differ over larger areas” explains the lack of division into subgroups. Maybe cut this phrase into two, and clarify the second part better.

Response: We have changed this sentence to make it easier to read. 

Discussion, page 36, lines 698 – 701: However, BBSP also generally has a more pelagic lifestyle [40,48] with larger foraging areas where stable isotope ratios differ over larger areas, and as such the feather isotopic signatures may vary less between individuals as they are averaged over a wider range of sources.

L614-616 – This sentence is a bit confusing to read. I would suggest replacing by “Indeed, year was only a significant factor in the WSP groups predicted to have moulting ranges extending further South (Table 2; Figure 1B and 5).”

Response: We have changed the structure of this sentence, following this suggestion. 

Discussion; Factors affecting moulting distribution; page 37, line 716 – 717: Indeed, year was only a significant factor in the WSP groups predicted to have moulting ranges extending further South (Table 5).

L629 – Add a space between “values” and “the”.

Response: We have changed this accordingly.

L678-681 - This sentence is too long, please add a comma after “origin areas for the northern species, as the (…)” or cut it in two sentences.

Response: We have included the suggested comma.

L751 – Add a comma and “the” in the sentence “migration strategies, spending the moulting”.

Response: We have changed the sentence accordingly.

Reviewer #3

This study by Ausems et al. measures stable isotope ratios of carbon, nitrogen and oxygen in feathers sampled from adults of multiple storm petrel species (European; Leach’s; black-bellied; Wilson’s) in the Northern and Southern hemispheres. Feathers are metabolically inert after synthesis and hence their isotopic composition reflects diet during synthesis. This is a well-established method in seabird ecology. The authors examined factors driving variation in feather δ13C and δ18O values and made predictions about moulting locations by using isoscapes. Overall the study is well designed and comprehensive; however, there are some issues that I believe the authors should address. 

Response: Thank you for the feedback. We are glad you are generally positive about our study, and we have considered your comments carefully. 

The Introduction would benefit from a clearer explanation of the use and interpretation of stable isotope ratios. How, for instance, should ecologists interpret stable isotope ratios of oxygen? These are not as common in the seabird literature as those of carbon and nitrogen. 

Response: We have added some extra sentences explaining the interpretation of δ18O values. 

Introduction, page 5, lines 106 – 111: Stable isotope compositions of both elements vary spatially in marine ecosystems; δ13C values are correlated with phytoplankton distribution [25,26] while δ18O values are correlated with salinity and fresh water input [27]. Therefore, they follow inshore/offshore gradients [26–28]. To our knowledge, this study is the first to combine δ13C and δ18O analyses to determine differences in moulting distributions of storm-petrels breeding sympatrically in both hemispheres.

Moreover, references to the isotopic niche in the Introduction are confusing given the lack of emphasis placed on δ15N values, which reflect an important niche axis. Please clarify and consider referencing some important literature on the isotopic niche (e.g. Newsome et al. 2007). 

Response: We did not put emphasis on δ15N on purpose. As δ15N is influenced by diet and location, and we do not have baseline data for the diet, we cannot use it to determine trophic niches or trophic levels for migrating storm-petrels. Stable nitrogen isotopic compositions serve as an important proxy for trophic level in the case of a local scale, e.g. during the breeding period when seabirds act as central place foragers having a restricted foraging range. During the non-breeding period pelagic seabirds roam freely through the vast oceans with spatially variable δ15N values. Predator trophic positions can only be inferred from bulk δ15N values if bulk δ15N values of lower trophic positions are known (Post, 2002; Weiss et al., 2009). We have included this explanation in the introduction.

Introduction, page 5, lines 111 – 117: Traditionally, δ13C is combined with δ15N to study species’ trophic and isotopic niches as nitrogen isotopic compositions serve as an important proxy for trophic level. However, this can only be used at local scales, e.g. during the breeding period when seabirds act as central place foragers having a restricted foraging range. During the non-breeding period pelagic seabirds roam freely through the vast oceans with spatially variable δ15N values. Specific predator trophic positions can only be inferred from bulk δ15N values if bulk δ15N values of lower trophic positions is known [29,30].

The Introduction may also benefit from some more specific information on the diets and foraging ecology of storm petrels. 

Response: This is relevant if we would have given δ15N a prominent place in our analyses, but we did not. Therefore, we decided to focus on moulting ecology and only included information on diet in the Materials and methods section.

Regarding the Materials and methods, the stable isotope analysis section requires greater precision; for instance, where is the explanation of the delta notation? 

Response: This section is following the minimum requirements suggested by IUPAC, see reporting guidelines for stable isotopes; a manifest by Coleman and Meier-Augenstein (2014). All standards, uncertainties and essential technical details are included. The delta notation is considered to be so widely used that it is not necessary to provide the equation.

The authors should also be careful not to refer to δ13C and δ18O values as elements. 

Response: We assume you meant we should make it clear we are talking about the stable isotopes, the instance in which it may not have been clear has been corrected. In other places we either use δ13C and δ18O notation or write ‘stable [element] isotope’. Both are correct.

I am not familiar with the conditional inference tree technique, but do the authors consider interactions among the various predictor variables of feather δ13C and δ18O values? 

Response: We did consider the interaction terms but decided against using the interaction δ13C and δ18O. The interaction between those two describes how they are related to each other but does not necessarily describe differences in the location of feather synthesis. As we were interested in localities, rather than how δ13C is related to δ18O, we decided not to use the interaction. 

Including interaction terms between predictors in the ctree formula is not meaningful because the models are robust to non-linearity, non-normality, multicollinearity and multiple interactions among explanatory variables (Johnstone et al., 2014; Quinn and Keough, 2002).

The authors may consider a separate heading for the ethics statement. 

Response: We have included an extra heading.

The opening paragraph of the Results is quite dense, and in my opinion would be better suited to a table. 

Response: We have included two more tables (Tables 1 and 3).

The authors might wish to consider sub-headings to organise the Discussion.

Response: We have included sub-headings in the discussion.

References used in the rebuttal letter

Arroyo, B., Mínguez, E., Palomares, L., Pinilla, J., 2004. The timing and pattern of moult of flight feathers of European storm-petrel Hydrobates pelagicus in Atlantic and Mediterranean breeding areas. Ardeola 51, 365–373.

Bearhop, S., Phillips, R.A., Mcgill, R., Cherel, Y., Dawson, D.A., Croxall, J.P., 2006. Stable isotopes indicate sex-specific and long-term individual foraging specialisation in diving seabirds 311, 157–164.

Ceia, F.R., Paiva, V.H., Fidalgo, V., Morais, L., Baeta, A., Crisóstomo, P., Mourato, E., Garthe, S., Marques, J.C., Ramos, J.A., 2014. Annual and seasonal consistency in the feeding ecology of an opportunistic species, the yellow-legged gull Larus michahellis. Mar. Ecol. Prog. Ser. 497, 273–284. https://doi.org/10.3354/meps10586

Ceia, F.R., Phillips, R.A., Ramos, J.A., Cherel, Y., Vieira, R.P., Richard, P., Xavier, J.C., 2012. Short- and long-term consistency in the foraging niche of wandering albatrosses. Mar. Biol. 159, 1581–1591. https://doi.org/10.1007/s00227-012-1946-1

Cherel, Y., Hobson, K.A., Weimerskirch, H., 2000. Using stable-isotope analysis of feathers to distinguish moulting and breeding origins of seabirds. Oecologia 122, 155–162.

Coleman, M., Meier-Augenstein, W., 2014. Ignoring IUPAC guidelines for measurement and reporting of stable isotope abundance values affects us all. Rapid Commun. Mass Spectrom. 28, 1953–1955. https://doi.org/10.1002/rcm.6971

Coplen, T.B., 2011. Guidelines and recommended terms for expression of stable-isotope-ratio and gas-ratio measurement results. Rapid Commun. Mass Spectrom. 25, 2538–2560. https://doi.org/10.1002/rcm.5129

Frith, R, Krug, D, Ronconi, R A, Wong, S N P, Mallory, M.L., Tranquilla, L.A.M., Frith, Rhyl, Krug, David, Ronconi, Robert A, Wong, Sarah N P, 2020. Diet of Leach’s storm-petrels (Hydrobates leucorhous) among three colonies in Atlantic Canada 27, 612–630.

Hobson, K.A., 2011. Isotopic ornithology: A perspective. J. Ornithol. 152. https://doi.org/10.1007/s10336-011-0653-x

Johnstone, C.P., Lill, A., Reina, R.D., 2014. Habitat loss, fragmentation and degradation effects on small mammals: Analysis with conditional inference tree statistical modelling. Biol. Conserv. 176, 80–98. https://doi.org/10.1016/j.biocon.2014.04.025

Post, D.M., 2002. Using stable isotopes to estimate trophic position: Models, methods, and assumptions. Ecology 83, 703–718. https://doi.org/10.1890/0012-9658(2002)083[0703:USITET]2.0.CO;2

Quinn, G.P., Keough, M.J., 2002. Experimental design and data analysis for biologists. Cambridge University Press.

Weiss, F., Furness, R.W., McGill, R.A.R., Strange, I.J., Masello, J.F., Quillfeldt, P., 2009. Trophic segregation of Falkland Islands seabirds: Insights from stable isotope analysis. Polar Biol. 32, 1753–1763. https://doi.org/10.1007/s00300-009-0674-6

---

## [Decision Letter · Decision Letter 1]

5 Jan 2021

PONE-D-20-32308R1

Birds of a feather moult together: differences in moulting distribution of four species of storm-petrels

PLOS ONE

Dear Dr. Ausems,

Thank you for submitting your manuscript to PLOS ONE. After careful consideration, we feel that it has merit but does not fully meet PLOS ONE’s publication criteria as it currently stands. Therefore, we invite you to submit a revised version of the manuscript that addresses the points raised during the review process.

We look forward to receiving your revised manuscript.

Kind regards,

Vitor Hugo Rodrigues Paiva, Ph.D.

Academic Editor

PLOS ONE

Reviewers' comments:

Reviewer's Responses to Questions

**Comments to the Author**

1. If the authors have adequately addressed your comments raised in a previous round of review and you feel that this manuscript is now acceptable for publication, you may indicate that here to bypass the “Comments to the Author” section, enter your conflict of interest statement in the “Confidential to Editor” section, and submit your "Accept" recommendation.

Reviewer #1: (No Response)

Reviewer #2: All comments have been addressed

Reviewer #3: All comments have been addressed

2. Is the manuscript technically sound, and do the data support the conclusions?

Reviewer #1: Yes

Reviewer #2: Yes

Reviewer #3: Yes

3. Has the statistical analysis been performed appropriately and rigorously? 

Reviewer #1: Yes

Reviewer #2: Yes

Reviewer #3: I Don't Know

4. Have the authors made all data underlying the findings in their manuscript fully available?

Reviewer #1: Yes

Reviewer #2: Yes

Reviewer #3: Yes

5. Is the manuscript presented in an intelligible fashion and written in standard English?

Reviewer #1: Yes

Reviewer #2: Yes

Reviewer #3: Yes

6. Review Comments to the Author

Reviewer #1: The revision of the manuscript “Birds of a feather moult together: differences in moulting range of four species of storm-petrels” from Ausmes and colleagues addressed most of the reviewers’ comments in a satisfactory manner and as a result, the manuscript has substantially improved (specially the results section). However, I still have a few minor comments.

Abstract

L35. I would specify: “to predict potential moulting areas of the sampled feather type”.

Introduction

L84-85. This sentence needs to be changed slightly. Maybe something like: “This latter species is considered the world’s most abundant seabird species. However, relatively little is known about storm-petrels ecology during the non-breeding period.”

L100-106. This is a nice and important addition to the manuscript from the previous version.

Materials and methods

The order in which the information is presented in this section is rather awkward and could use some re-ordering. The authors mentioned number of birds captured and feather collection before the “data collection” section. The field site locations are described outside of the “field study” section.

L229. Introduce acronyms the first time they are used.

Reviewer #2: I think the manuscript was greatly improved by all the recommendations and tweaks that the authors incorporated from the reviewers. Now the manuscript is clearer, comprehensive from the beginning to the end, and the authors defended very well the rebuttal. I only have some minor things to point:

L26: I cannot make you use the new nomenclature of the LSP, but I still think you should use it – if it is changing, the old one is meant to stop being used. I would suggest using it right from the abstract on, and further down, instead of “been changed to Hydrobates leucorhous” you should write “described before as Oceanodroma leucorhoa”.

L191: Add “(PCR)” after “polymerase chain reaction” since it is a very well-known acronym.

L195-196: “(…) we tested the sex of a total of 4 BBSP (2 females; 1 male)”. It sums to 3 birds, not 4.

L223: Uniform the values (0.1 and 0.5 or 0.10 and 0.50)

L433 and L436: You have empty parenthesis, please complete with what was supposed to come in them.

L646: “factors ( Fig 5).” Delete the space between parenthesis and Fig.

Reviewer #3: (No Response)

7. PLOS authors have the option to publish the peer review history of their article (what does this mean?). If published, this will include your full peer review and any attached files.

Reviewer #1: No

Reviewer #2: **Yes: **Ana Rita Carreiro

Reviewer #3: No

---

## [Author Response · Author response to Decision Letter 1]

6 Jan 2021

Vitor Hugo Rodrigues Paiva 06.01.2021

Academic Editor

PLOS ONE

Dear Dr. Vitor Paiva,

On behalf of all authors, I would like to thank you and the reviewers for the thorough review process and the valuable comments on our manuscript. As after the first round of reviews, we have addressed all comments raised by the reviewers below. We have started each answer with “Response”, and included the section, page and line numbers referring to the revised manuscript with changes marked when appropriate.

We hope that our changes are satisfactory and meet the expectations of the PLOS ONE editors and reviewers. Please, do not hesitate to contact me if you have any questions, comments or would like further clarification on anything. Thank you for your time considering our manuscript.

Yours sincerely,

Anne N.M.A. Ausems

Reviewer #1

The revision of the manuscript “Birds of a feather moult together: differences in moulting range of four species of storm-petrels” from Ausmes and colleagues addressed most of the reviewers’ comments in a satisfactory manner and as a result, the manuscript has substantially improved (specially the results section). However, I still have a few minor comments.

Response: Thank you for the positive words! 

Abstract

L35. I would specify: “to predict potential moulting areas of the sampled feather type”.

Response: We have changed the sentence accordingly.

Abstract; page 2; lines 34 – 35: […] data to predict potential moulting areas of the sampled feather type.

Introduction

L84-85. This sentence needs to be changed slightly. Maybe something like: “This latter species is considered the world’s most abundant seabird species. However, relatively little is known about storm-petrels ecology during the non-breeding period.”

Response: We have changed the sentence as suggested.

Introduction; page 4; lines 85 – 87: The latter species is considered the world’s most abundant seabird species. However, relatively little is known about storm-petrel ecology during the non-breeding period.

L100-106. This is a nice and important addition to the manuscript from the previous version.

Response: We are pleased to hear that.

Materials and methods

The order in which the information is presented in this section is rather awkward and could use some re-ordering. The authors mentioned number of birds captured and feather collection before the “data collection” section. The field site locations are described outside of the “field study” section.

Response: To address the concerns raised here, we have changed the section header to “Study species and location”, specified that we captured adults and moved the last paragraph of this section to the next (i.e. under Field Study).

Materials and methods; Study species and location; page 6; lines 129 – 132: 

Study species and location

We captured ESP and LSP adults in August of 2018 (n = 52; n = 56, respectively) and 2019 (n = 40; n = 37, respectively) on the island of Mykines, Faroe Islands (62°05´N, 07°39´W), and BBSP and WSP adults during the austral summer of 2017 (n = 15; n = 100, respectively) and 2018 (n = 19; n = 126, respectively).

Materials and methods; Data collection; Field study; page 8; lines 186 – 193: It took several weeks for the sampled rectrices to be fully grown (ESP 30.6 ± 8.5 d; LSP 40.8 ± 13.0 d; BBSP 18.5 ± 2.5 d; WSP 18.7 ± 3.0 d; [59]). Thus, the rectrix formation period, overlapping to a considerable extent with flight feather moult[14,24], includes a considerable part of the non-breeding period, even if feather growth started at the end of the breeding season. Although sampling tail-feathers increases the chance of sampling a feather moulted during the breeding period in LSP, we considered the uncertainty around the location of the start of tail moult too great to justify adding the negative effect of increasing feather gaps by sampling a more central feather.

L229. Introduce acronyms the first time they are used.

Response: We have removed this part of the sentence and now refer to ‘the analyses’ in general. 

Materials and methods; Statistical analyses; page 11; lines 239 - 240: The results for the analyses including the outliers are reported in the supplementary data (S2).

 

Reviewer #2

I think the manuscript was greatly improved by all the recommendations and tweaks that the authors incorporated from the reviewers. Now the manuscript is clearer, comprehensive from the beginning to the end, and the authors defended very well the rebuttal. I only have some minor things to point:

Response: Thank you for the positive feedback!

L26: I cannot make you use the new nomenclature of the LSP, but I still think you should use it – if it is changing, the old one is meant to stop being used. I would suggest using it right from the abstract on, and further down, instead of “been changed to Hydrobates leucorhous” you should write “described before as Oceanodroma leucorhoa”.

Response: We have followed your advice, and changed the species name in the abstract and introduction to Hydrobates leucorhous, and changed the sentence in the materials and methods to mention the old name. 

Materials and methods; Study species and location; pages 6 – 7; lines 138 – 140: The species name of LSP was therefore recently changed by BirdLife from Oceanodroma leucorhoa to Hydrobates leucorhous [43] though the old nomenclature is still widely used as well.

L191: Add “(PCR)” after “polymerase chain reaction” since it is a very well-known acronym.

Response: We have included “(PCR)” as suggested.

L195-196: “(…) we tested the sex of a total of 4 BBSP (2 females; 1 male)”. It sums to 3 birds, not 4.

Response: You are correct; the absolute total is four but only three were sexed based on the PCR results. We have corrected it. 

L223: Uniform the values (0.1 and 0.5 or 0.10 and 0.50)

Response: We have added a 0 after 0.5 to make the values uniform. 

L433 and L436: You have empty parenthesis, please complete with what was supposed to come in them.

Response: These were unfortunately left behind after revising the section. They are removed now.

L646: “factors ( Fig 5).” Delete the space between parenthesis and Fig.

Response: We have corrected this accordingly.

 

Reviewer #3 

(No Response)

---

## [Editor Report · Decision Letter 2]

7 Jan 2021

Birds of a feather moult together: differences in moulting distribution of four species of storm-petrels

PONE-D-20-32308R2

Dear Dr. Ausems,

We’re pleased to inform you that your manuscript has been judged scientifically suitable for publication and will be formally accepted for publication once it meets all outstanding technical requirements.

Kind regards,

Vitor Hugo Rodrigues Paiva, Ph.D.

Academic Editor

PLOS ONE
---

## [Editor Report · Acceptance letter]

11 Jan 2021

PONE-D-20-32308R2 

Birds of a feather moult together: differences in moulting distribution of four species of storm-petrels 

Dear Dr. Ausems:

I'm pleased to inform you that your manuscript has been deemed suitable for publication in PLOS ONE. Congratulations! Your manuscript is now with our production department. 

Kind regards, 

on behalf of

Dr. Vitor Hugo Rodrigues Paiva 

Academic Editor

PLOS ONE